# Programmable on-chip nonlinear photonics

Ryotatsu Yanagimoto[1,2✉], Benjamin A. Ash[1], Mandar M. Sohoni[1], Martin M. Stein[1,5], Yiqi Zhao[1], Federico Presutti[1,6], Marc Jankowski[2,3], Logan G. Wright[1,2,5], Tatsuhiro Onodera[1,2] & Peter L. McMahon[1,4✉]

Nonlinear optics[1] plays a central role in many photonic technologies, both classical[2–5] and quantum[6–8]. However, the function of a nonlinear-optical device is typically determined during design and fixed during fabrication[9], restricting the use of nonlinear optics to scenarios in which this inflexibility is tolerable. Here we present a photonic device with highly programmable nonlinear functionality: an optical slab waveguide with an arbitrarily reconfigurable two-dimensional distribution of $\chi^{(2)}$ nonlinearity. The nonlinearity is realized using electric-field-induced $\chi^{(2)}$ (refs. 10–16), and the programmability is engineered by massively parallel control of the electric-field distribution within the device using a photoconductive layer and optical programming with a spatial light pattern. To showcase the versatility of our device, we demonstrate spectral, spatial and spatio-spectral engineering of second-harmonic generation by tailoring arbitrary quasi-phase-matching grating structures[1] in two dimensions. The programmability of the device makes it possible to perform inverse design of grating structures in situ, as well as real-time feedback to compensate for fluctuations in operating and environmental conditions. Our work shows that we can break from the conventional one-device–one-function paradigm, potentially expanding the applications of nonlinear optics to situations in which fast device reconfigurability is desirable—such as in programmable optical quantum gates and quantum light sources[7,17–19], all-optical signal processing[20], optical computation[21] and adaptive structured light for sensing[22–24].

Nonlinear optics (NLO)[1] encompasses a diverse range of processes, including sum-frequency, difference-frequency and parametric generation, and has become a central tool in many optical technologies[2–8]. The NLO processes used in nonlinear photonics are usually not accessible in raw materials because, in their natural state, raw materials do not satisfy the conditions for phase-matching, which is essential for an NLO process to be efficient[1]. A particularly efficacious and flexible technique for achieving phase-matching artificially is quasi-phase matching (QPM). In QPM, a periodic spatial modulation of $\chi^{(2)}$ optical nonlinearity, which is referred to as a QPM grating, compensates for phase mismatch among interacting light waves[25,26]. A simple QPM grating enables highly efficient coherent-wave mixing, whereas more complicated ones can realize highly nontrivial nonlinear-optical functions. Many exotic QPM grating structures have been explored, enabling a wide range of functions including high-harmonic generation[27,28], arbitrary pulse shaping[29], quantum-pulse gates[17] and holographic generation of structured light[30–32].

These functions of an NLO device, including engineering phase-matching, often require special structures to be 'sculpted' in the raw nonlinear-optical material, and this is usually done by means of nanofabrication processes[9]. For instance, a QPM grating can be formed by periodically inverting the crystal axis of a nonlinear material. Fabricating such a structure requires sophisticated techniques—for example, epitaxial growth of semiconductors on orientation-patterned wafers[33]

or domain inversion of ferroelectric materials by means of lithographically patterned electrodes[34]. Although this sculpting-based model has driven decades of progress in nonlinear photonics, it restricts the design and use of photonic devices because each device is typically optimized for one function that is fixed when the device is made. The performance of the device can also be very sensitive to fabrication imperfections as well as operating and environmental conditions that deviate from those that the device was designed for[35], which lowers the yield of correctly functioning devices.

We present an approach that avoids many of these disadvantages: NLO based on a programmable nonlinear waveguide. Our proposed device is a planar optical waveguide whose two-dimensional distribution of $\chi^{(2)}$ optical nonlinearity, $\chi^{(2)}(x, z)$, can be arbitrarily programmed, that is, dynamically set and updated ($x$ and $z$ are the transverse and longitudinal dimensions of the waveguide, respectively; see Fig. 1a). Such programmable $\chi^{(2)}$ nonlinearity allows flexible engineering of QPM gratings to perform various NLO functions with a single device. The programmable $\chi^{(2)}$ nonlinearity is induced by biasing the $\chi^{(3)}$ nonlinearity with an electric field $E_{bias}(x, z)$, leading to $\chi^{(2)}(x, z) = 3\chi^{(3)}E_{bias}(x, z)$. Electric-field-induced $\chi^{(2)}$ nonlinearity[36–38] has recently been used to engineer a variety of NLO processes[10–16] in which the bias field was applied by means of lithographically patterned electrodes[10,11] or all-optically[12–16]. The all-optical approach reconfigures the spatial pattern of nonlinearity depending on how the device is optically pumped.

[1]School of Applied and Engineering Physics, Cornell University, Ithaca, NY, USA. [2]Physics & Informatics Laboratories, NTT Research, Inc., Sunnyvale, CA, USA. [3]E. L. Ginzton Laboratory, Stanford University, Stanford, CA, USA. [4]Kavli Institute at Cornell for Nanoscale Science, Cornell University, Ithaca, NY, USA. [5]Present address: Department of Applied Physics, Yale University, New Haven, CT, USA. [6]Present address: Research Laboratory of Electronics, Massachusetts Institute of Technology, Cambridge, MA, USA. ✉e-mail: ryotatsu.yanagimoto@ntt-research.com; pmcmahon@cornell.edu

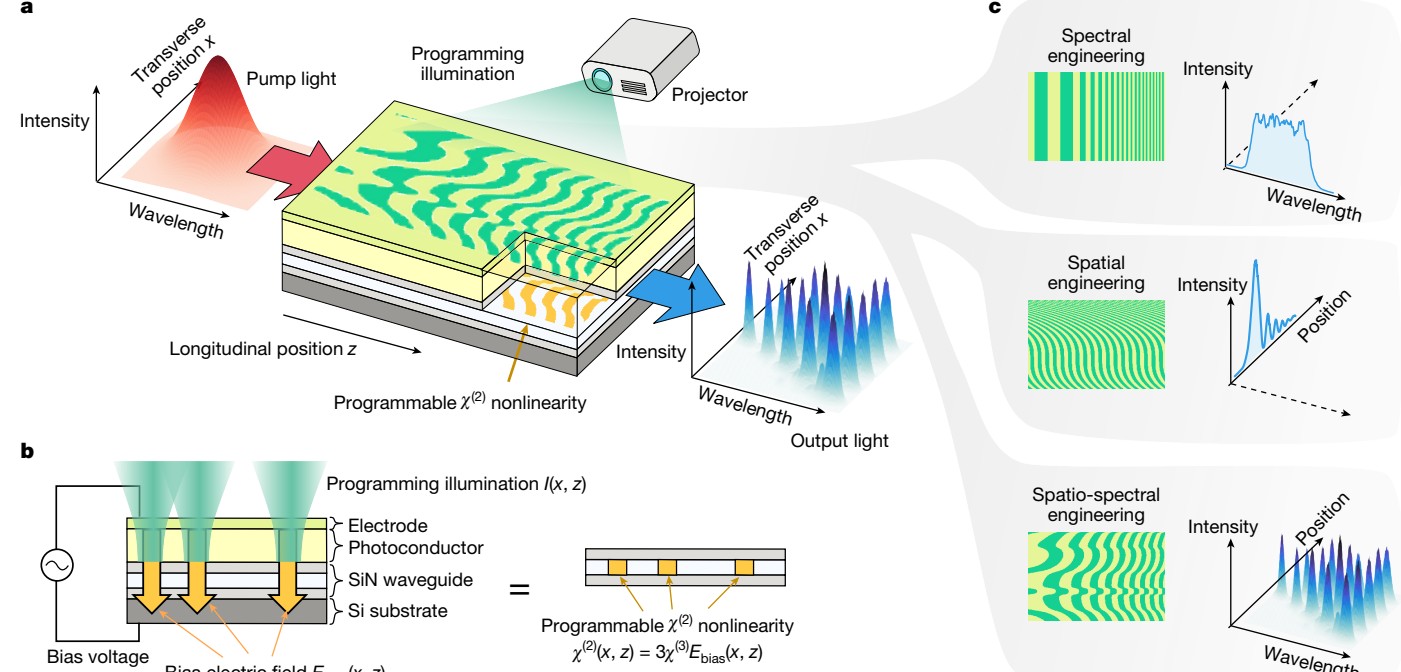

**Fig. 1 | Illustration of the working principle and capabilities of a programmable nonlinear waveguide. a**, Structured light projected on the surface of a planar waveguide plays the role of a programming illumination $I(x, z)$, inducing the same pattern of $\chi^{(2)}$ nonlinearity, $\chi^{(2)}(x, z)$, which allows versatile control of broadband SHG by means of QPM. Here $x$ and $z$ are the transverse and longitudinal positions on the waveguide, respectively. **b**, The structure and physical working mechanism of a programmable waveguide. The device is composed of a SiN waveguide (2.05-µm-thick SiN core and 1-µm-thick SiO$_2$ cladding at the top and bottom), a photoconductor layer (7.5-µm-thick SRN) and a transparent electrode (20-nm-thick ITO). The photoconductor,

when illuminated by green (532-nm) laser light, becomes locally conductive, letting the external bias electric field through to the waveguide core. The resulting spatially shaped $E_{bias}(x, z)$ induces a spatially programmable $\chi^{(2)}$ nonlinearity according to $\chi^{(2)}(x, z) = 3\chi^{(3)}E_{bias}(x, z)$. See Methods and Supplementary Information for details on the device fabrication and electrical characterization of the device, respectively. **c**, Varying the longitudinal and transverse structure of QPM gratings enables spectral and spatial control, respectively, of NLO. By programming the full two-dimensional structure of $\chi^{(2)}(x, z)$, we can simultaneously engineer the spectral and spatial structure of the generated output light.

However, completely arbitrary spatial patterns of $\chi^{(2)}$ nonlinearity have not been realized, and this challenge also applies to other approaches to tuning $\chi^{(2)}$ nonlinearity, such as using ferroelectric nematic liquid crystals[39,40]. Reconfigurable QPM structures have also been limited to one-dimensional geometries.

In our planar waveguide device, we use lithography-free photoconductive electrodes and patterned optical illumination to program arbitrary spatial patterns into the bias field, $E_{bias}(x, z)$. This programmable bias field produces a corresponding programmable $\chi^{(2)}(x, z)$. Patterned optical illumination has previously been used to program the real[41] and imaginary[42] parts of the refractive-index distribution of planar waveguides. Here we demonstrate a programmable $\chi^{(2)}(x, z)$ nonlinearity with a dynamic range (that is, contrast) of 0.47 pm V$^{-1}$, a spatial resolution of 7.5 µm and a functional area ($z \times x$) of approximately 0.7 × 0.4 cm, with updates possible every second (Supplementary Information). Using this full two-dimensional programmability, we experimentally realized complex QPM structures and demonstrated flexible control over the spectral, spatial and spatio-spectral dynamics of broadband second-harmonic generation (SHG). Moreover, the real-time reconfigurability of the device enables in situ inverse design and optimization of QPM grating structures, allowing us to engineer very unusual optical spectral and spatial shapes in a way that is robust to experimental imperfections.

## Design and operating principle of the device

Our programmable nonlinear waveguide and how we realized arbitrary two-dimensional distributions of nonlinearity $\chi^{(2)}(x, z)$ is illustrated in Fig. 1. Our programmable nonlinear waveguide comprised several

layers (Fig. 1b; see Methods). The waveguide was made on a conductive silicon substrate. On top of the substrate was a silicon nitride (SiN) optical waveguide comprising silicon dioxide (SiO$_2$) cladding layers and a SiN core layer. On top of the upper cladding layer was a layer of photoconductive material—silicon-rich silicon nitride (SRN). Finally, a transparent electrode was deposited on the photoconductor layer. During operation, a bias electric field was applied across the entire stack by connecting a voltage source to the substrate and the top electrode.

To realize a programmable $\chi^{(2)}(x, z)$, we shone programming illumination with a spatial intensity pattern $I(x, z)$ onto the top of the device. The photoconductor layer became conductive where light intensity was highest and let the electric field from the bias voltage through to the SiN core layer (Supplementary Information). Consequently, the pattern of the programming illumination $I(x, z)$ resulted in a pattern of the bias field $E_{bias}(x, z)$ inside the core. The third-order nonlinear polarization induced by the sum of the bias field $E_{bias}$ and the optical field $E_{opt}$ (the field inside the waveguide, travelling in the $z$ direction; not the programming illumination field) can be expanded as

$$P_{NL} = \chi^{(3)}(E_{bias} + E_{opt})^3 = \chi^{(3)}(E_{opt}^3 + 3E_{bias}E_{opt}^2 + 3E_{bias}^2E_{opt} + E_{bias}^3). \quad (1)$$

Here an effective quadratic nonlinearity arises as a term proportional to $E_{opt}^2$, whose coefficient $\chi^{(2)}(x, z) = 3\chi^{(3)}E_{bias}(x, z)$ is proportional to the bias field. The central operating principle of our device can be summarized as: patterned programming illumination $I(x, z)$ on the photoconductor layer induces a spatial pattern of electric field $E_{bias}(x, z)$ inside the waveguide core, which then induces a spatial pattern of optical nonlinearity $\chi^{(2)}(x, z)$.

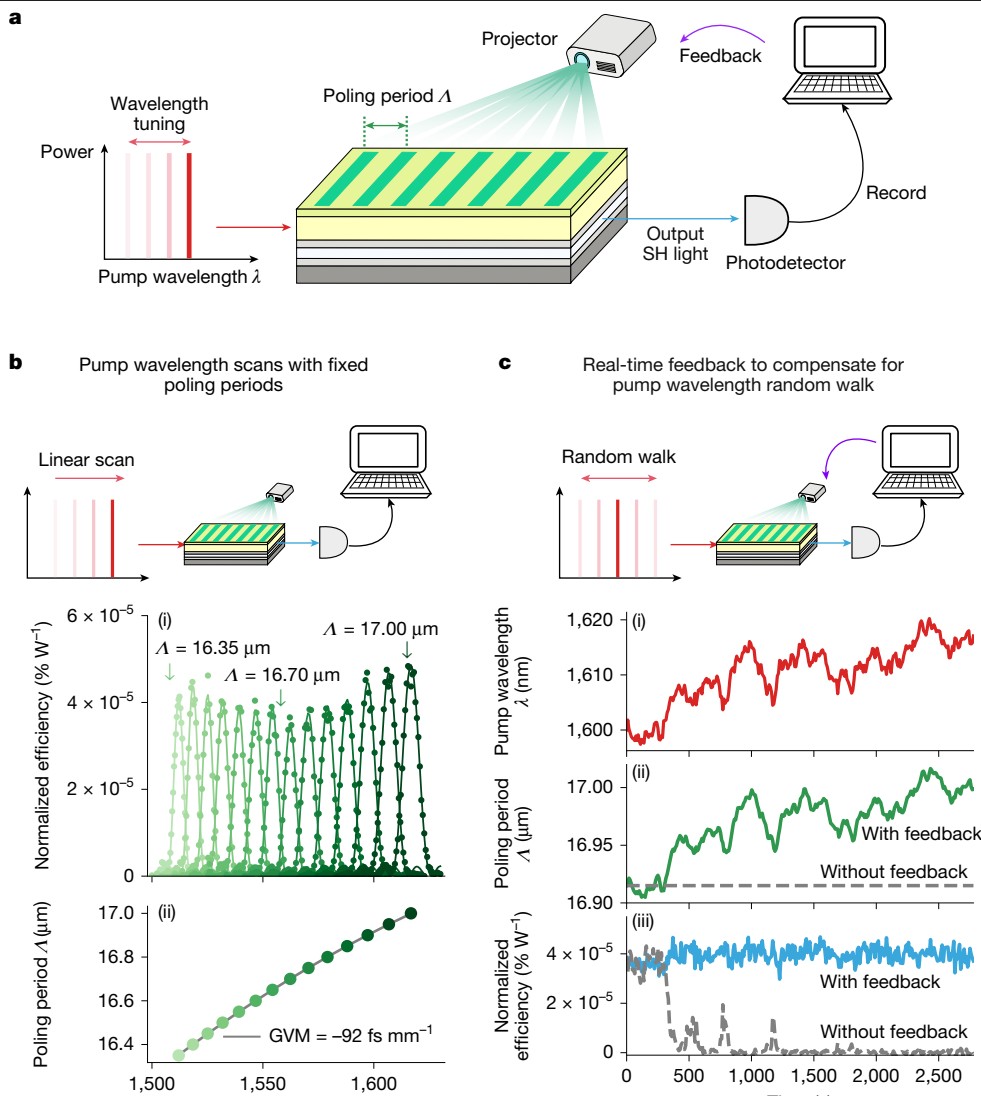

**Fig. 2 | Real-time programmable periodic poling with a programmable nonlinear waveguide. a**, Experimental set-up. We pumped a prototype programmable nonlinear waveguide using a CW laser with a tunable wavelength $\lambda$. A grating pattern with period $\Lambda$ was projected on the waveguide, realizing QPM for a SHG process. The output SH power was measured by a photodetector and the measurements could be used to update $\Lambda$. **b**, Nonlinear-optical characterization of the device. (i) For various choices of $\Lambda$, we scanned $\lambda$ and measured the SHG conversion efficiency, which we report as an efficiency normalized by input power. (ii) The optimal pump wavelength $\lambda$ for each poling period $\Lambda$. The quadratic fit between $\Lambda$ and an optimal $\lambda$ yielded a nominal poling period of 16.65 μm and group-velocity mismatch (GVM) of −92 fs mm⁻¹ between

the fundamental and second harmonic at 1,560 nm. The colours of the markers serve as legends for $\Lambda$ in (i). **c**, Real-time feedback control of $\Lambda$ to compensate for a random walk of $\lambda$ shown in (i). To compensate for these fluctuations, we dithered $\Lambda$ and used the measured SHG signal to update $\Lambda$ in a way that maximizes the signal. The evolution of $\Lambda$ is shown as a solid green line in (ii). In (iii), the SHG efficiency with and without such real-time feedback control is shown as blue solid and grey dashed lines, respectively. For all of the measurements, the bias voltage was 1,000 V. The nominal on-chip pump power, inferred from the transmission, ranged between 2.2 and 4.5 mW, depending on the wavelength, primarily because of the wavelength-dependent loss of the core. See Supplementary Information for experimental details.

The specific design we used for our device (choice of materials and layer thicknesses) constrained the $\chi^{(2)}(x, z)$ we could realize (Supplementary Information). The first constraint was that fringing of the electric fields blurred the mapping from $I(x, z)$ to $E_{bias}(x, z)$, limiting the smallest programmable feature size to about 7.5 μm. The second constraint was that the resistor–capacitor (RC) time constant of the device limits the update speed of $\chi^{(2)}(x, z)$ to about 20 Hz. The slow speed of the projector used in the set-up limited the update speed even more, to about 1 Hz.

The nonlinear-optical functionality of our device can be tailored by modifying the structure of $\chi^{(2)}(x, z)$ (Fig. 1c). For example, consider narrowband SHG, in which light at frequency $\omega_1$ is converted to light at frequency $\omega_2 = 2\omega_1$. Efficient frequency conversion requires that

the momenta of interacting waves be matched. This requirement is quantified by the native phase mismatch,

$$\Delta k = k_2 - 2k_1, \qquad (2)$$

in which $k_j$ is the wavenumber of light at frequency $\omega_j$ ($j \in \{1, 2\}$). Efficient SHG can be achieved when this phase mismatch is compensated by a QPM grating—that is, by a periodic modulation of $\chi^{(2)}$ with a period designed to offset $\Delta k$, for example, $\chi^{(2)}(x, z) \approx \sin(\Delta k z)$ (ref. 1).

We performed experiments demonstrating three types of use of our device. First, we engineered the longitudinal (that is, $z$) structure of $\chi^{(2)}(x, z)$ to control which wavelengths interact efficiently, enabling spectral-domain engineering of NLO. Second, we tailored the transverse (that is, $x$) structure of $\chi^{(2)}(x, z)$, enabling spatial-domain engineering.

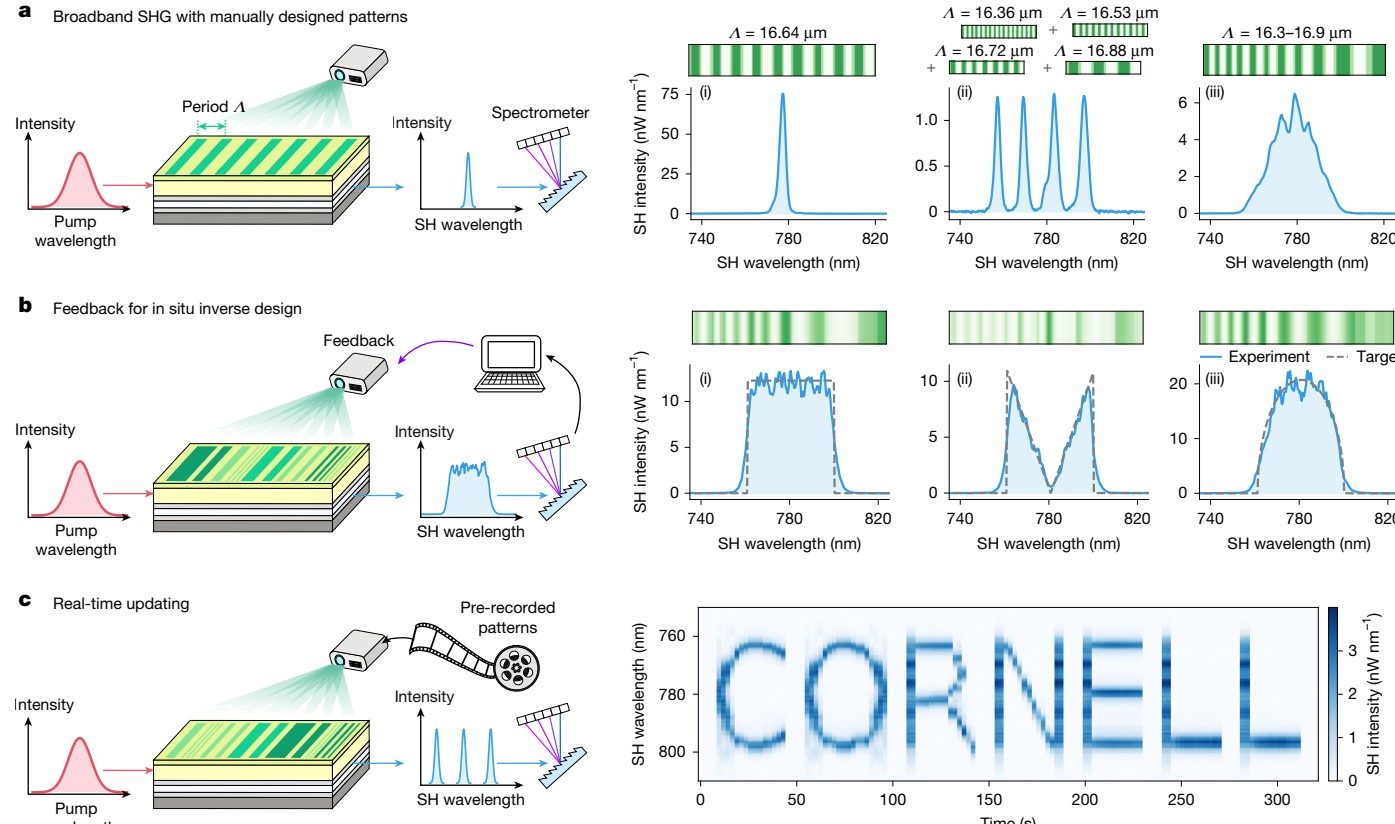

**Fig. 3 | Spectral engineering of SHG. a**, Output SH spectrum of broadband SHG pumped by ultrashort pump pulses for various illumination patterns. The bias voltage was 500 V. (i) Periodic grating with a period $\Lambda$ = 16.64 μm. (ii) Superposition of four grating patterns with different periods. (iii) An adiabatically chirped grating pattern. Owing to the rapid spatial oscillations of these QPM gratings, showing the raw illumination patterns is not visually informative. Instead, in the green patterns shown above the results plots, we present the QPM grating patterns downsampled to a spatial period of 17 μm in the longitudinal direction. The same applies to the patterns shown in **b**. See Supplementary Information for the original (non-downsampled) illumination patterns. **b**, By constructing a feedback loop based on the measured SH spectrum, we optimized the illumination pattern to obtain various target SH spectra. Dashed lines represent the target spectrum. The bias voltage was 800 V. **c**, The illumination pattern was updated in real time to output a sequence of SH spectra, using pre-recorded illumination patterns. We show the results for drawing 'CORNELL' in the SH spectrum, with time as the horizontal axis of the image. The bias voltage was 800 V. See Supplementary Information for experimental details. For all of the measurements, we used a pulse laser with 60-fs pulse duration and 100-MHz repetition rate. The average on-chip pump power inferred from the transmission was 6 mW.

For instance, if we set $\chi^{(2)}(x, z) \approx \sin(\Delta kz + \phi(x))$, with a spatially varying phase term $\phi(x)$, the generated second-harmonic (SH) light acquires a corresponding spatial phase profile, $e^{i\phi(x)}$, thereby shaping the output field in the transverse ($x$) direction. Finally, we used the full two-dimensional programmability of $\chi^{(2)}(x, z)$—in both the longitudinal and transverse directions—enabling simultaneous control of the light in both the spectral and spatial domains, giving rise to spatio-spectral engineering. Below, we present the results of our experiments.

## Real-time programmable periodic poling

First we characterized the basic nonlinear-optical properties of the device by programming canonical QPM gratings with different poling periods $\Lambda$ and measuring the power of the SHG when the device was pumped with a continuous-wave (CW) laser that had a tunable wavelength between $\lambda$ = 1,500 and 1,630 nm (Fig. 2a). For this initial characterization, we fixed the poling period $\Lambda$ and scanned the wavelength of the pump laser to measure the SHG conversion efficiency. Depending on the value of $\Lambda$, different wavelengths of pump light undergo phase-matched SHG (Fig. 2b). The nonlinearity of the device is proportional to the bias electric field $E_{bias}$, which should not be set higher than the breakdown field of the material. With the highest $E_{bias}$ we applied, we found the induced $\chi^{(2)}$ nonlinearity of $\chi^{(2)}$ = 0.47 pm V$^{-1}$ (Supplementary Information).

We took advantage of the ability to reprogram the poling period in our initial device characterization, but in these experiments, the programming did not have to take place quickly. To showcase the ability of our device to be programmed in real time, we performed an experiment to show that it is possible to compensate for environmental noise and drifts by adjusting the poling period on a timescale of approximately 1 s (Fig. 2c). To emulate large noise, we artificially modulated the pump wavelength $\lambda$ so that it followed a Gaussian random walk. The compensation task was to dynamically change the poling period $\Lambda$ to maximize the SHG efficiency, without being given information about the random changes in $\lambda$. We used a feedback scheme in which $\Lambda$ was dithered to obtain an error signal and the error signal was used to update $\Lambda$. The data clearly show that, when the feedback controller was on, $\Lambda$ closely followed the evolution of the pump wavelength $\lambda$, maintaining a high level of SHG efficiency. On the other hand, when the feedback controller was off, the SHG efficiency dropped to near zero relatively quickly.

## Spectral engineering

Next we show how the programmable nonlinear waveguide can be used to manipulate the spectral shape of the generated SH light by programming $\chi^{(2)}(x, z)$ in the longitudinal ($z$) direction. In each of the experiments, we pumped a prototype device with an ultrashort pulse laser and measured the output SH spectrum using a spectrometer.

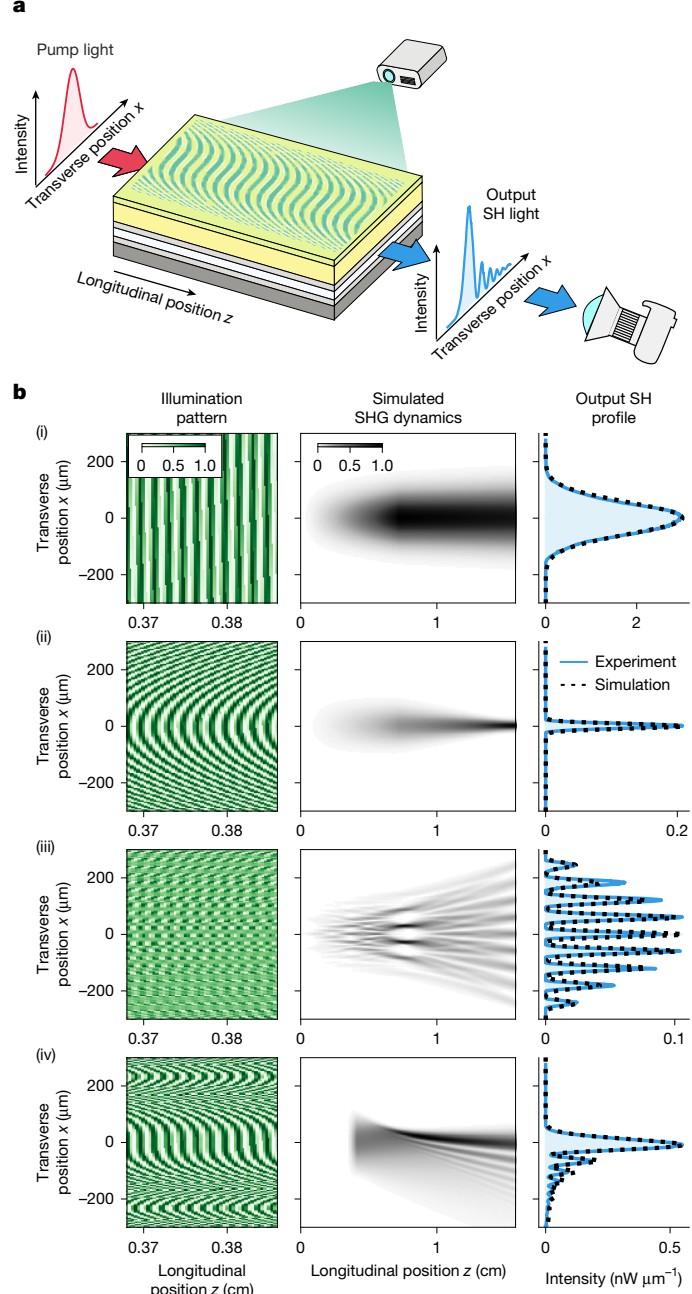

**Fig. 4 | Spatial engineering of SHG. a**, Experimental set-up. The waveguide was pumped with a pulse pump laser with fixed Gaussian spatial profile, and the spatial distribution of the generated SH on a camera was measured. **b**, A part of the normalized programming illumination pattern (left column), simulated SHG dynamics within the waveguide (middle column) and a comparison between the experimentally measured and simulated SH profiles (right column). (i) Monotonic grating pattern. (ii) Quadratically chirped grating pattern. (iii) Superposition of nine quadratically chirped grating patterns with transverse offsets. (iv) Cubically chirped grating pattern. For all of the measurements, we used a pulse laser with 60-fs pulse duration and 100-MHz repetition rate. The average on-chip pump power inferred from the transmission was 6 mW. The bias voltage was 600 V. See Supplementary Information for experimental details.

We measured the SHG spectra for various manually designed QPM grating structures (Fig. 3a) to verify that our device could reproduce well-known results in NLO. As a reference case, we programmed a grating pattern with a single period $\Lambda$, which phase-matched SHG for a particular pump wavelength (Fig. 3a (i)). This manifested as a single, narrow peak in the SH spectrum. Beyond such a simple grating pattern, we programmed a summation of several grating patterns with different periods (Fig. 3a (ii)). This super-grating pattern can simultaneously phase-match various SHG processes and can generate several wavelengths of SH, which are visible as peaks in the recorded spectrum. Finally, we programmed a chirped grating—a grating in which the period is changed as a function of the longitudinal position—and observed broadband SHG output (Fig. 3a (iii)), consistent with previous non-programmable demonstrations of adiabatic SHG[43]. For more general spectral features, refs. 44,45 presented and demonstrated frameworks for analytically designing QPM grating structures, which could be used to design patterns for the programmable device we present.

Up to this point, the illumination patterns we used to program the waveguide were designed manually, in that we designed them on the basis of standard knowledge of NLO. To demonstrate the ability to shape the SHG spectrum in ways that are probably impractical using conventional NLO devices, which do not support real-time reconfiguration, we arbitrarily shaped and dynamically updated the SHG spectrum. We achieved this by constructing a real-time feedback loop between the broadband SH spectral measurement and update of the programming illumination patterns (Fig. 3b) (Supplementary Information). This approach enables spectral engineering in a way that is robust against imperfections and miscalibrations in the device and experimental set-up.

The illumination patterns that are optimized for in inverse design can be stored and later retrieved to program a sequence of nonlinearity distributions in real time. We demonstrated this by drawing 'CORNELL' in the SH spectrum as a function of time (Fig. 3c); the programming illumination pattern was updated every few seconds, for which the update speed was limited by that of the spatial light modulator (about 1 s) we used to pattern the illumination.

## Spatial engineering

In this section, we show the ability to engineer the spatial structure of light generated using our programmable nonlinear waveguide by controlling phase-matching conditions in the transverse dimension, as has been demonstrated previously in non-programmable NLO[30,32,46,47]. Here we pumped the programmable nonlinear waveguide with a pulse with a fixed Gaussian spatial beam shape (beam waist: 132 μm) and imaged the output SHG profile for various programming illumination patterns, that is, different distributions of $\chi^{(2)}(x, z)$ (Fig. 4a).

As a reference, we first projected a simple, flat (that is, constant in the transverse dimension) grating pattern with a period of 16.75 μm, corresponding to phase-matched SHG near 790 nm. The output SHG also had a Gaussian profile with a large beam waist of 94 μm (Fig. 4b (i)). Then we performed experiments in which the phase of the QPM grating was spatially varied and observed that the generated SH light inherited the phase of the grating, which allowed us to engineer the spatial profiles of the SHG. For instance, by quadratically chirping the phase of the grating, we were able to focus the generated SH light to the output facet, resulting in a substantially narrower beam waist of 16 μm (Fig. 4b (ii)).

More complex patterns can be produced by superimposing several grating structures; for example, nine quadratically curved grating patterns, evenly spaced in the transverse direction, focused SHG into nine distinct peaks (Fig. 4b (iii)). This approach can, in principle, be used to generate arbitrary superpositions of Gaussian peaks.

Diffraction-free beams that maintain their spatial profiles during propagation are used in microscopy and imaging[47]. NLO can generate Airy beams—one-dimensional non-diffracting beams—by applying a cubic chirp to a QPM grating in the transverse direction. We reproduced the seminal demonstration from ref. 30 with our programmable platform: our spatially resolved measurement of the waveguide output

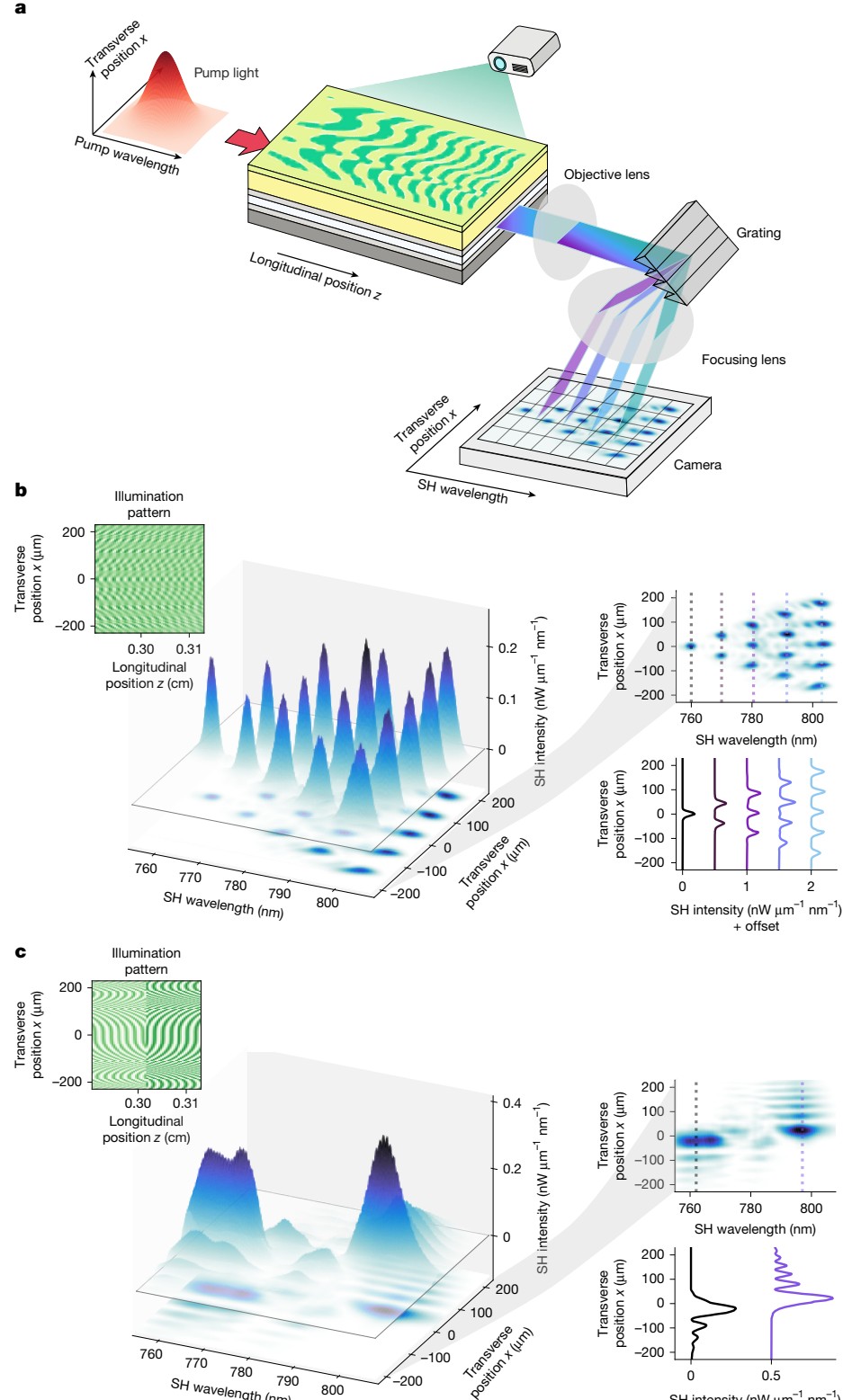

**Fig. 5 | Spatio-spectral engineering of SHG. a**, Experimental set-up. We combined a reflective grating with a 4*f* imaging set-up to record spectrally resolved one-dimensional spatial profiles of the output SH light. The waveguide was pumped by pulses with a fixed Gaussian spatial profile. **b**, Results for an illumination pattern designed to generate various numbers of spatial peaks at five different wavelengths. **c**, Results for an illumination pattern designed to generate oppositely chirped Airy beams at two different wavelengths. In each

of **b** and **c**, the left inset shows a part of the projected grating pattern. The bottom-right inset shows the spatial distribution of the SH light at various wavelengths, marked with dashed lines in the top-right inset. For all of the measurements, we used a pulse laser with 60-fs pulse duration and 100-MHz repetition rate. The average on-chip pump power inferred from the transmission was 40 mW. The bias voltage was 600 V. See Supplementary Information for experimental details.

(Fig. 4b (iv)) clearly shows the characteristic asymmetric interference fringes of an Airy beam.

The SH output profiles we recorded are in excellent agreement with theoretical simulations across all measurements; the simulations used only a single fitting parameter, for the overall amplitude. The match between experiment and theory makes it possible to optimize illumination patterns for engineering spatial features entirely in silico.

## Spatio-spectral engineering

So far, we have shown independent control of spectral and spatial features of SHG by tailoring the longitudinal and transverse structure of QPM gratings, respectively. In this section, we show that it is possible to use the full two-dimensional programmability of the $\chi^{(2)}$ nonlinearity to simultaneously tailor the spatial and spectral profiles of the generated light. The experimental set-up is illustrated in Fig. 5a. We projected patterns of light onto the programmable waveguide and pumped it with broadband optical pulses. The output SH light was measured by spectrally resolved one-dimensional imaging, in which the spectrum was recorded for each transverse spatial position by using a diffraction grating and a camera.

We aimed to obtain a spatio-spectral hologram in which the SHG output has a spatial profile that is a function of the output wavelength. To do this, we superimposed various QPM grating patterns with different longitudinal periods. In our first spatio-spectral experiment, we designed the grating structure to generate one, two, three, four and five spatial peaks at five different wavelengths (Fig. 5b). As shown in the one-dimensional hyperspectral image captured by the camera, we observed clearly separated Gaussian peaks localized in both space and wavelength. In our second experiment, we took inspiration from the SHG-based hologram proposed in ref. 30 as a means to generate different Airy beams for different wavelengths. Here we show that oppositely chirped Airy beams can be generated by combining two grating patterns with different longitudinal periods and opposite cubic spatial chirps. We clearly observed the characteristic asymmetric interference fringes of the Airy beams but in opposite directions for two separate wavelengths (Fig. 5c).

## Discussion and outlook

### Summary of the results

We developed a programmable nonlinear waveguide with an arbitrarily reconfigurable two-dimensional distribution of $\chi^{(2)}$ nonlinearity. By engineering QPM gratings—both conventional and exotic, inverse-designed ones—we demonstrated versatile control over broadband SHG across the spectral, spatial and spatio-spectral domains. The programmability of our device enabled real-time in situ optimization of QPM grating structures using feedback from experimental measurements and the real-time playback of pre-designed gratings. Notably, all of the results reported in this paper were obtained using a single programmable nonlinear waveguide design and the same pulse laser (except in Fig. 2, in which we used a tunable CW laser). This highlights the flexibility and multifunctionality of programmable nonlinear waveguides.

### Limitations and potential for improvements

In this study, we demonstrated a broad range of functions using programmable nonlinearity in a planar waveguide. However, our experimental prototype has several practical drawbacks that, although not fundamental, need to be addressed if the programmable-waveguide approach is to be widely used. First and foremost is the weak optical nonlinearity. On our prototype device, we estimated the maximum programmable $\chi^{(2)}$ nonlinearity to be 0.47 pm V$^{-1}$. Although this lies on the higher side of the reported values for electric-field-induced $\chi^{(2)}$ in SiN nanophotonics (0.03–0.50 pm V$^{-1}$)[12–15], it is still low compared with conventional nonlinear-optical materials[6]. Fortunately, there are known methods to increase this value, improve the device speed

and eliminate the need for AC operation (Supplementary Information). Second, we observed relatively large optical loss, ranging from a nominal value of about 1 dB cm$^{-1}$ for wavelengths >1,550 nm to a peak value of roughly 5 dB cm$^{-1}$ around 1,520 nm, owing to residual hydrogen in plasma-enhanced chemical vapour deposition (PECVD)-deposited SiN. This limits the use of the device for wavelengths near 1,520 nm. High-temperature annealing is known to substantially reduce such absorption by driving out hydrogen impurities, and optical loss as low as 0.4 dB m$^{-1}$ has been demonstrated in SiN nanophotonic devices fabricated with low-pressure chemical vapour deposition[48,49]. Third, the conversion efficiency of the device was lower than that of other demonstrations in SiN nanophotonics, primarily because of the weak field confinement of the planar waveguide geometry[12–15]. To verify that it is possible to apply our approach to nanophotonic structures with tight field confinement, we fabricated and tested a programmable nonlinear channel waveguide. As shown in Methods, we observed a normalized efficiency of $\eta_{norm} = 2 \times 10^{-3}$% W$^{-1}$, representing a 40-fold improvement over our planar waveguide result of $\eta_{norm} = 5 \times 10^{-5}$% W$^{-1}$. Forming a resonant structure with a waveguide could further boost the conversion efficiency by recirculating light. Assuming state-of-the-art loss values for SiN and the performance of the channel waveguide we demonstrated, we estimate that a programmable microring resonator could possibly reach $\eta_{norm} > 1 \times 10^{7}$% W$^{-1}$, which exceeds the present state of the art in integrated platforms (Supplementary Information).

## Prospective applications

The ability to realize arbitrary $\chi^{(2)}(x, z)$ distributions makes our device platform very versatile, particularly in enabling devices that must seamlessly switch between several functions. As shown in the demonstration of the programmable channel waveguide (Methods), our approach is, in principle, compatible with a wide range of device geometries beyond planar waveguides and could be seamlessly co-integrated with conventional photonic systems. Concretely, we have quantitatively analysed the potential for our approach to be applied in four example application areas: (1) on-chip arbitrary pulse shapers; (2) reconfigurable quantum frequency converters; (3) widely wavelength-tunable integrated light sources; and (4) quantum light sources with programmable entanglement structure (Supplementary Information). For these applications, we find that future SiN programmable nonlinear photonics could potentially achieve competitive or even state-of-art performance assuming the $\chi^{(2)}$ nonlinearity demonstrated in this work, all while allowing the flexibility enabled by programmability. In several of our application analyses, a crucial feature enabling programmable SiN to deliver competitive performance is the exceptionally low material loss of SiN, which enables snaking channel waveguides to be metres long and still have low loss[49,50]. The long path lengths possible in SiN can fully compensate for the low $\chi^{(2)}$ nonlinear coefficient and increase the programmable degrees of freedom. We expect that these four applications are not the only future possibilities: more speculatively, all-optical signal processing for optical communications[20] could benefit from reconfigurable nonlinear processes, as could classical optical computation[21] and sensing with structured light[22–24]. The scope of applications could further expand with the use of materials with higher inducible $\chi^{(2)}$ nonlinearities[10]. Finally, in situ inverse design may enable quantitative control over nonlinear-optical processes for which we do not have accurate simulation models at present.

In conclusion, the ability to programmably control nonlinearity has the potential to circumvent the limitations of the conventional one-device–one-function paradigm. The programmable nonlinear waveguide we have proposed and the demonstrations of reconfigurable SHG we have reported with a prototype device take a step into this new frontier of NLO.

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

## Methods

### Fabrication of programmable nonlinear planar waveguides

In this section, we describe the fabrication process for the programmable nonlinear planar waveguide. As illustrated in Extended Data Fig. 1a, the device was composed of a stack of several material layers. Silicon Valley Microelectronics (SVM) supplied the substrates, including the bottom cladding and the core layer. The substrate was a conductive, boron-doped Si wafer with a resistivity in the range 0.01–0.02 Ω cm. The bottom cladding consisted of a 1-μm-thick wet thermal oxide layer, onto which approximately 2 μm of low-stress SiN was deposited by means of PECVD. Using a Metricon prism coupler, we measured the film thickness as $d_{core} = 2.05$ μm, with a thickness variation of approximately 50 nm across a 4-inch wafer. The refractive index of the film was specified as 1.98 at a wavelength of 632.8 nm.

We performed rapid thermal annealing (RTA) on the wafers acquired from SVM at 650 °C for 3 min. This RTA process reduced the refractive index of the film and eliminated undesired fluorescence in the near-infrared region when the waveguide was pumped near 800 nm. Because the results presented in the main text did not depend on this pumping wavelength, the RTA process shifted the phase-matching conditions without causing notable adverse effects.

Next we deposited a 1-μm-thick layer of $SiO_2$ as the top cladding using our in-house PECVD system (Oxford Instruments PlasmaPro 100 PECVD), thereby forming the planar SiN waveguide. To make the waveguide programmable, we further deposited a 7.5-μm-thick layer of SRN using PECVD. The SRN was deposited at an RF power of 200 W with gas flows of $SiH_4$: 8 sccm, $H_2$: 40 sccm and $N_2$: 2,000 sccm. We note that no $NH_3$ was used.

At this stage, we cleaved the wafer into rectangular pieces of approximately $1.0 \times 1.5$ cm. Although cleaving typically produces facets that are sufficiently clean for light coupling, more polishing can further improve the beam profile quality. Finally, we deposited a 20-nm-thick layer of indium tin oxide (ITO) as a transparent electrode using a physical vapour deposition (PVD) system (PVD 75, Kurt J. Lesker). It was important to leave several millimetres of space between the electrode and the chip edge to prevent electrical breakdown of the air at the boundary. A picture of the resultant programmable nonlinear planar waveguide is shown in Extended Data Fig. 1b.

### Fabrication and characterization of programmable nonlinear channel waveguides

In the main text, we primarily focus on a programmable planar waveguide. Although this design allowed us to engineer spatial features of light, it came with the drawback of lower light confinement, which reduced the conversion efficiency. One possible solution to this challenge is to fabricate a channel waveguide structure to prevent spatial diffraction, which enables light propagation over longer distances while maintaining tight transverse confinement. To demonstrate the compatibility of our approach with this channel waveguide geometry, we fabricated a prototypical programmable channel waveguide using SiN and characterized its performance.

The device was fabricated with the following steps. First, SVM provided a Si substrate with a 1-μm bottom oxide cladding and a 2-μm PECVD-grown SiN layer, identical to the substrate type used for the programmable planar waveguides. Next we deposited a 500-nm-thick $SiO_2$ layer using PECVD, followed by sputtering a 200-nm-thick Cr layer. For photolithography, we spin-coated a deep ultraviolet (DUV) anti-reflective coating and photoresist. Exposure was performed using a DUV stepper (PAS 5500, ASML), after which we developed the photoresist and wet-etched the Cr to form a hard mask. We then etched through the $SiO_2$ and SiN layer using $CHF_3/O_2/N_2$ gases in a plasma etcher (PlasmaPro 100 RIE, Oxford Instruments). We then deposited 3 μm of $SiO_2$ with PECVD and subsequently etched away 2.5 μm of oxide using $CHF_3/O_2$ gases in the same plasma etcher, which

added 500 nm of oxide on top of the waveguide. Notably, 1.5 μm of oxide remained on the waveguide sidewalls owing to the conformal oxide deposition and anisotropic oxide etching. This extra oxide on the sidewall reduces loss into the photoconductor layer. At this stage, we performed RTA at 800 °C for 5 mins. Finally, we deposited a 12-μm layer of SRN and sputtered a 20-nm layer of ITO as a transparent electrode. A cross-sectional scanning electron microscope (SEM) image of the fabricated channel waveguide is shown in Extended Data Fig. 2a. The SiN core was approximately 4 μm wide and 2 μm high. The waveguide had an overall length of 1.5 cm, with the programmable region limited to approximately 7 mm owing to constraints of the imaging set-up.

We then conducted a nonlinear-optical characterization of the device. We coupled CW pump light with various wavelengths between 1,540 and 1,600 nm to the waveguide. The pump light is polarized in the vertical (that is, $y$) direction, which is expected to mainly excite the fundamental transverse magnetic (TM) mode, and the SHG takes place to the fundamental TM mode of the SH wavelength. We show the mode profiles of these modes in Extended Data Fig. 2b. We measured the generated SHG while scanning the period of the monotonic QPM grating. We calculated the normalized SHG conversion efficiency by dividing the detected SHG power by the pump power squared measured after the waveguide, including appropriate corrections for propagation losses and collection inefficiencies (Supplementary Information). The measurement results are shown in Extended Data Fig. 3, which clearly demonstrate the capability to achieve phase-matching at a desired pump wavelength by adjusting the QPM grating period.

Note that our waveguide core measures approximately $2 \times 4$ μm in cross-section, supporting several spatial modes even at the pump wavelength. Moreover, the skewed aspect ratio of the core made it difficult to couple the pump light exclusively into the fundamental mode. Consequently, the pump power measured after the waveguide could include contributions from higher-order modes that do not contribute to SHG, which explains the wavelength-dependent variations in the normalized conversion efficiency. Thus, the efficiencies shown in Extended Data Fig. 3 should be considered a lower bound of the device performance. The maximum normalized conversion efficiency observed was $\eta_{norm} = 2 \times 10^{-3}\%$ $W^{-1}$, corresponding to a normalized slope conversion efficiency of $\eta_0 = 4 \times 10^{-3}\%$ $W^{-1}$ $cm^{-2}$, showing an approximately 40-fold improvement over the programmable nonlinear planar waveguide results presented in the main text. We note that reducing the core thickness below the micron scale to achieve single-mode operation causes the first-order poling period to fall below the smallest programmable feature size. If single-mode operation is given priority, higher-order QPM can be used, albeit with reduced conversion efficiency. At the same time, the tighter field confinement increases nonlinear coupling and could compensate for the reduction of conversion efficiency.

## Data availability

Experimental data and scripts to replicate the figures in this paper are available at https://doi.org/10.5281/zenodo.17074707 (ref. 51).

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

**Acknowledgements** We thank T. Chervy, J. Clark, N. Flemens, R. Hamerly, O. Jaramillo, G. Marega, J. Moses, R. Nehra, E. Ng, T. Takahashi, T. Wang, A. Willner and Y. Yamamoto for helpful comments and discussions. We thank C. Bacon and Springer Nature Author Services for English language editing. We thank NTT Research for their financial and technical support. We gratefully acknowledge the Air Force Office of Scientific Research for funding under award number FA9550-22-1-0378 and the National Science Foundation for funding under award number CCF-1918549. This work was performed in part at the Cornell NanoScale Facility,

a member of the National Nanotechnology Coordinated Infrastructure (NNCI), which is supported by the National Science Foundation (grant NNCI-2025233). P.L.M. gratefully acknowledges financial support from a David and Lucile Packard Foundation Fellowship.

**Author contributions** R.Y., T.O., L.G.W. and P.L.M. conceived the project. R.Y., B.A.A., M. M. Stein, T.O. and L.G.W. designed the devices. R.Y., T.O. and P.L.M. designed the experiments. R.Y., B.A.A. and Y.Z. fabricated the device, with aid and recipe development from M. M. Stein and T.O. R.Y. and M. M. Sohoni designed and built the imaging set-up to program the $\chi^{(2)}$ nonlinearity pattern. R.Y. designed and built the set-up for nonlinear-optical experiments, with aid from F.P. and M.J. R.Y. and M. M. Sohoni wrote the code for real-time optimization of the QPM grating. R.Y. performed the experiments, analysed the results and produced the figures. R.Y. and P.L.M. wrote the manuscript, with input from all authors. P.L.M. supervised the project.

**Competing interests** M. M. Stein, L.G.W., T.O. and P.L.M. are listed as inventors on a patent application (WO2023220401A1) on 2D-programmable waveguides.

**Additional information**
**Correspondence and requests for materials** should be addressed to Ryotatsu Yanagimoto or Peter L. McMahon.

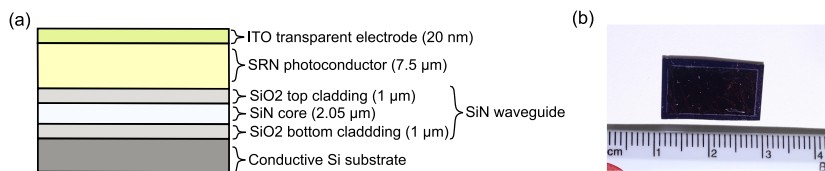

(a)
ITO transparent electrode (20 nm)
SRN photoconductor (7.5 μm)
SiO2 top cladding (1 μm)
SiN core (2.05 μm) — SiN waveguide
SiO2 bottom cladding (1 μm)
Conductive Si substrate

(b)

**Extended Data Fig. 1 | Structure of a programmable nonlinear planar waveguide. a**, Illustration of the stack structure of a programmable nonlinear planar waveguide. **b**, Photograph of a programmable nonlinear planar waveguide that was fabricated and used to produce parts of the results in the main text. The rectangular line visible on the surface of the chip is the edge of the transparent electrode.

(a)

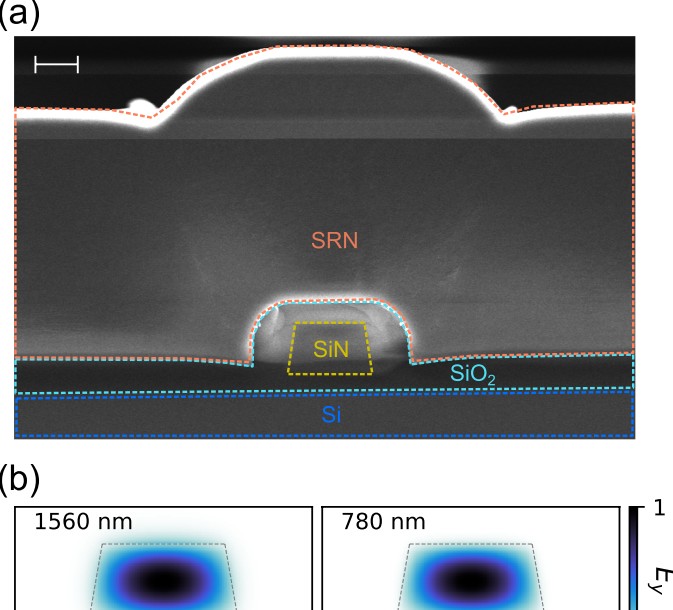

SRN

SiN

SiO$_2$

Si

(b)

1560 nm          780 nm

$E_y$

**Extended Data Fig. 2 | Structure of a programmable nonlinear channel waveguide. a**, SEM image of the cross-section of a programmable nonlinear channel waveguide. We highlight the boundaries between different materials with dashed lines. Scale bar, 2 µm. **b**, Numerically simulated distribution of the vertical electric field $E_y$ of the fundamental TM mode at wavelengths of 1,560 nm (left) and 780 nm (right). Scale bar, 1 µm. The simulation was performed with EMpy (ref. 52).

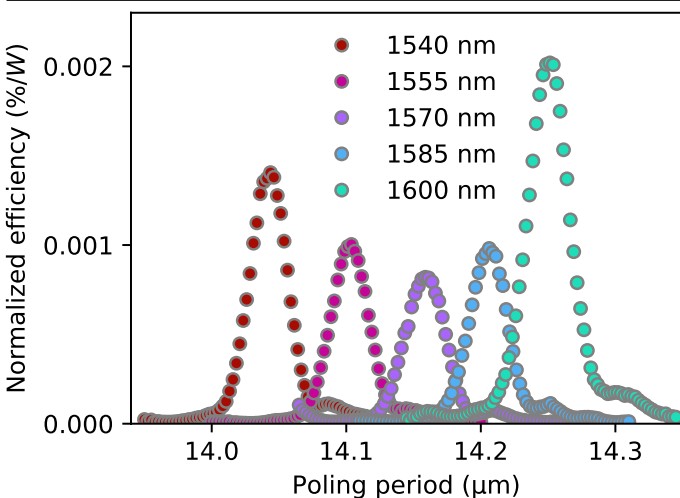

**Extended Data Fig. 3 | Tunable SHG on a programmable nonlinear channel waveguide.** Normalized SHG conversion efficiency of the programmable nonlinear channel waveguide for various pump wavelengths and programmed QPM grating periods. Approximately 4 mW of pump power was detected after the waveguide. We used a bias voltage of 1,500 V with frequency of 10 Hz.