## [Peer Review File · Nature]

Programmable on-chip nonlinear photonics

Corresponding Author: Dr Ryotatsu Yanagimoto

Version 0:

Reviewer comments:

Referee #1

(Remarks to the Author)

This is an interesting paper in which the authors present a method to program the nonlinearity of a slab waveguide by illuminating it with different light patterns. The waveguide material is SiN, that exhibits third order nonlinearity, and the effective second order nonlinearity is achieved by external biased voltage. The illumination locally changes the conductivity of a photon-conductive layer on top of the waveguide, thereby generating a patterned bias field, and therefore a modulated effective second order nonlinearity. The ability to program the nonlinearity is demonstrated in a series of experiments, including quasi-phase-matched (QPM) second harmonic generation with different QPM periods, engineering the spectral, the spatial or the spatio-spectral properties of the generated light, feedback loop to track variations of the pump wavelength and more.

Overall, I believe that this is a significant new result, with important potential applications. QPM processes are widely used nowadays in many applications, but to large extent, the properties of the nonlinear crystal are fixed at the fabrication process and cannot be altered. This paper presents a practical method to obtain a programmable nonlinear crystal. The parameters of the crystal (effective nonlinearity of 0.47 pm/V) are still about an order of magnitude lower than those of periodically poled ferroelectrics, but the authors discuss in the supplementary methods to further increase it.

I therefore believe that this paper can be considered for publication in Nature after properly addressing the following points:

1. Most of the experimental results are given in arbitrary units. What was the pump power in mW? and what was the measured second harmonic power in mW? This info should be provided.
2. Many of the reported measurements cover the range of 1550-1570 nm, but only at figure S.10 of the supplementary, they report huge losses in this range. This point should be made much clearer in the paper itself. In fact, it means that the effective wavelength range of this device is quite limited at this point.
3. The authors present in Section IV an iterative method to shape the spectrum of the generated light. It should be noted that this is not essential, since there is an analytical method to design the phase matching spectrum, see for example Shiloh et al, Optics Letters 37, 3591 (2012); Leshem et al, Optics Letters 39, 5370 (2014).
4. In the supplementary section, Fig. S3, the authors suggest a method to improve the performance using a photoconductive core. Wouldn't such a device suffer from very high losses? Moreover, these losses will be dependent on the QPM pattern.
5. Can this method be used with a channel waveguide, instead of a planar waveguide? This can potentially provide much higher conversion efficiency.

Referee #2

(Remarks to the Author)

I co-reviewed this manuscript with one of the reviewers who provided the listed reports.

Referee #3

(Remarks to the Author)

The authors demonstrate a versatile approach to reconfigurable nonlinear frequency conversion based on the effective $\chi^{(2)}$ nonlinearity realized through the DC-Kerr effect. Because the DC field can be electronically reconfigured in space and

time, this allows the effective $\chi(2)$ nonlinearity to be reconfigured. The authors demonstrate programmable quasi-phase matching for second harmonic generation, broadband and inverse-designed SHG profiles (i.e., spectral engineering), control of the spatial profile of the SHG light (spatial engineering), and combined spatio-spectral engineering.

To realize such re-configurability of the DC electric field, the authors develop a clever approach that does not require any fine fabrication, but instead relies on a single unpatterned electrode and a photoconductive layer that, when illuminated by a structured field, results in a structured DC field that reaches the waveguide core.

Overall, I found this work to be creative and original, and the manuscript is well-written and easy to follow. The results are intuitive, pleasing, and of high quality. In contrast to the many nonlinear photonics works which optimize performance for a limited set of scenarios, here the authors maximize flexibility and re-configurability. They use an extremely simple device platform, with most of the complexity shifted to the SLM-based optical projector system. This approach doesn't really compete with nonlinear integrated photonics devices, where the point is to limit off-chip resources, but is perhaps more suited for applications in which periodically-poled crystals that are part of a free-space optical system are used.

This brings me to my main reservation for publication in Nature, which is that I don't know what problem(s) this work actually solves. The observed efficiency of $5 \times 10^{-5} \% / W$ is very low (8 orders of magnitude lower than the highest efficiencies shown with the DC-Kerr effect in SiN and around 5 orders of magnitude lower than even inefficient poled crystals), which renders it hard to use in almost any application, I believe. Supplement S1 details how the magnitude of the $\chi(2)$ nonlinearity can be improved by a factor of 50x through changes in materials, etc, but even with this, the efficiencies would appear to be very low. And the applications the authors have claimed are all described very qualitatively, without any hard metrics outlined that could help a reader understand if the approach described in this paper can be truly relevant if the advances in Supplement S1 are realized. I believe that the authors need to more thoroughly address which applications their approach is particularly well-suited for, the level of performance needed for that application (or at least, what is currently available), and how they can get there, for this work to rise to the level of Nature. While it is true that not all Nature papers need clear applications, given that there aren't really any new physical principles being explored and that the work is essentially a technology demonstration, I think the above is necessary.

Finally, the manuscript comes with a 38-page supplement which, on the one hand, is great in terms of providing lots of details, but on the other hand is too much to expect most readers to go through. I request that the authors carefully consider what material is included in the supplement and ensure that their principal results and conclusions are sufficiently addressed in the main text, with the supplement used for backup/details but not any primary material. I also think that they need a table of contents and introductory section for the supplementary material so that it is easier for readers to navigate.

Referee #4

(Remarks to the Author)

Yanagimoto et al. report a fully programmable on-chip nonlinear photonic slab waveguide system capable of arbitrary reconfiguration through two-dimensional distribution of $\chi(2)$ nonlinearity. This is achieved by patterned illumination $I(x,z)$ on a photoconductive layer, which induces a spatial electric field pattern within the waveguide core, thereby creating a 2D distribution of optical nonlinearity $\chi(2)(x,z)$. As the authors fairly note, this work combines electric-field-induced $\chi(2)$ (as in Refs. 12–18) with patterned illumination techniques (Refs. 44–45). While the mechanism of electric-field-induced $\chi(2)$ has been well studied in previous years, a device with such strong programmable nonlinearity has not been demonstrated before. I believe this is a highly interesting and practically significant advancement in nonlinear photonic devices with high programmability.

Programmable linear optics has been well developed for classical and quantum photonic applications, but programmable nonlinearity remains relatively unexplored (previously, nonlinear crystals/sources were typically fixed, with reconfigurable pumping being the primary tuning method, e.g., Nat. Photonics 17, 573–581 (2023), which very differs from reconfiguring the nonlinear process itself). A relevant paper (Nat. Photon. 2025, <https://doi.org/10.1038/s41566-025-01660-x>) was recently published, but it employs a very different approach to tuning nonlinearity. This work stands out due to its conceptual novelty and technological improvements, particularly in reducing linear loss in the electric-field-induced process and achieving substantial $\chi(2)$. The authors also provide a detailed discussion on further enhancing $\chi(2)$ to reach levels comparable to lithium niobate, addressing key factors such as bandgap wavelength (critical for SHG and SPDC), pattern resolution, and optical loss.

Suggestions for improvement before publication:

1. Clarification of $\chi(2)(x,z) = 3\chi(3)E_{\text{bias}}(x,z)$ is necessary. The derivation of this relationship is not entirely clear in the current manuscript. Given its importance in understanding SHG in third-order SiN materials, the authors should elaborate on whether the factor of three is dependent on the waveguide structures or materials.

2. The authors demonstrate flexible and comprehensive control over SHG dynamics, including spectral, spatial, and spatio-spectral (even real-time) tuning. A discussion on possible applications would strengthen the manuscript. For instance, static spectral and spatial control of QPM is highly relevant for engineering SPDC-based quantum light sources, enabling high-purity, indistinguishable single photons and multimode squeezed light (see conventional SPDC crystals from USTC's Pan group and waveguide sources from Paderborn's Silberhorn group). Additional insights into real-time dynamic control of QPM or nonlinearity could help envision future applications.

3. The authors mention that the applied electric field E_{bias} must remain below the material's breakdown threshold. However, potential breakdown due to the external pump light and illumination light on the photoconductive layer and in the core layer should also be addressed. Another critical aspect is the lifetime of the patterned $\chi(2)$ distribution. How stable is it after applying the electric field and illumination? Its resilience under environmental factors (e.g., temperature fluctuations) should be clarified.

4. The current slab waveguide features a $2\mu\text{m}$ SiN core with $1\mu\text{m}$ SiO₂ cladding, supporting multimodes at both pump and SHG wavelengths. Could the authors discuss the feasibility of reducing the core thickness to achieve single-mode operation (vertically)? Etched ridge waveguides, which confine light to a single mode in both vertical and horizontal directions, are more relevant for photonic integration. Would the proposed nonlinear photonic waveguide be adaptable to such single-mode, 1D reconfigurable structures? This is of significant practical importance.

Related comment: The authors note the sensitivity of 1D PPLN structures to fabrication variations, necessitating adaptive poling. However, the primary reason for PPLN sensitivity lies in thin-film LN layers (hundreds of nanometers thick); thicker layers are far less susceptible (see conventional bulk PPLN waveguides). The authors should clarify their statement.

Could the authors also comment on how the current (or improved) nonlinear waveguides might be integrated with conventional photonic waveguides for system-level applications?

5. Minor corrections : The color scheme for SiO₂ cladding is inconsistent between Fig. 1a (gray) and Fig. 1b/Fig.6 (pink), which may cause confusion. The x-axis label is missing in Fig. 4a.

This work pioneers a new direction in nonlinear optics and photonic devices, representing a milestone achievement in the field. With the above points addressed, I would be pleased to recommend its publication in Nature.

Version 1:

Reviewer comments:

Referee #1

(Remarks to the Author)

The authors have made significant revisions to their paper, following the reviewers comments. These include some clarifications and comparisons to other works, adding some missing data, in particular related to the experimental measurements, report of a new result of channel waveguide and 4 detailed proposals for future applications of this technology. Overall, the authors have made a serious effort to address the reviewers comments and I believe that the manuscript was improved in this process.

Perhaps the most important new result is the report on realization and characterization of a channel waveguide. This enabled to address one of the weak points of this new technology - its relatively low nonlinear conversion efficiency. Now that they use channel waveguide, they report a significant increase of 40 times in conversion efficiency.

As for the proposals for future applications with this technology, I think that they are interesting, but some of the numbers there are very optimistic. As an example, for Potential Application 1: Highly programmable on-chip optical pulse shaper and for Potential Application 2: Programmable quantum frequency converter, the authors assume that they can maintain the efficiency they obtained for a length of 0.7 mm over a propagation distance of 1.5 meters. It is not clear whether they can maintain the exact phase matching condition over such a long distance. This is a major issue with other, more mature waveguide technologies, for example using thin film LiNbO₃. In addition, the current loss level they reported is 0.4 dB/cm, so it still requires dramatic reduction in the loss values in order to have interaction lengths of meters.

Nevertheless, I still think that this paper represents a major new result in nonlinear optics, and I recommend accepting it to Nature.

Reviewer name: Ady Arie

Referee #2

(Remarks to the Author)

I co-reviewed this manuscript with one of the reviewers who provided the listed reports.

Referee #3

(Remarks to the Author)

In my previous review, I noted the creativity and quality of the results presented in this manuscript but had concerns about whether the low conversion efficiency would be prohibitive for practical applications, and also wanted more clarity as to envisioned applications of the authors' programmable nonlinear waveguide approach. I am pleased to say that in their revised manuscript, supplementary material, and response letter, the authors have done a thorough job of addressing both concerns. I therefore endorse publication in Nature.

For the former, they have first carefully outlined sources of inefficiency, focusing on how their effective second-order nonlinearity is actually quite significant (e.g., on par or larger than the highest demonstrated in SiN) and it is mostly their specific device geometry and limitations with loss and modal confinement that prevent higher efficiency from being obtained. They then conducted new experiments in a channel waveguide geometry, introducing lateral confinement that is absent in the prior slab waveguide results, and show that this improves the SHG efficiency by about a factor of 40, making it about one order of magnitude lower than prior results in channel waveguides that exhibit a greater degree of confinement (in those works, the effective $\chi^{(2)}$ is photoinduced). Given that there are many possible ways to further engineer the system, in terms of stronger optical mode confinement, reduced optical loss, resonant enhancement, etc, and none of these are fundamentally incompatible with the authors' approach (albeit there will likely be tradeoffs), I believe their demonstrations have now answered the basic question of whether useful conversion efficiencies can be realized with their approach.

With regards to the latter, the authors have presented four future applications in the supplementary material (summarized in the main text), all of which I believe are credible and point to what may be achievable with some additional levels of device improvement.

My remaining comments are below:

1. The authors could consider including Table R1 in their supplementary material. This clear comparison against the prior literature can be helpful for readers. They could potentially additionally include a 'bandwidth' column since conversion efficiency at a single frequency isn't the only metric in many important cases.
2. I didn't really understand what I was supposed to be looking at in Figure 6(b) – is the channel waveguide visible in this optical image, or is the main point to show the waveguide length? What is the rectangular region within the chip denoting?
3. In the channel waveguide experiments, did the authors observe any SHG in absence of an explicit applied DC field (i.e., do they observe any SHG mediated by the photogalvanic effect for DC field creation)? It could be interesting to compare the results of a photo-induced SHG vs. SHG via an explicit DC field if that is the case.
4. Do the authors have any ideas about the fundamental and second harmonic mode identities (e.g., polarization and transverse profile) in the channel waveguide experiments? If so, it would be useful to note this and potentially including simulated field profiles in Fig. 7. Regarding polarization, it would seem that their DC field is vertically oriented – does that mean they focus primarily on TM-polarized modes?
5. I found the description of the fabrication process for the channel waveguides to be a little unclear in a couple of places:
 - a. They mention first depositing 500 nm thick SiO₂ on the wafer they ordered (which had a lower SiO₂ cladding a SiN core), then performing photolithography and pattern transfer to a Cr hard mask, and then etching through the SiN layer. Just to confirm, aren't they etching through both the 500 nm SiO₂ as well as the 2 μ m SiN in this step?
 - b. They mention then depositing an additional 3 μ m of SiO₂ before etching away 2.5 μ m of it. One may wonder why they didn't simply deposit 500 nm of additional SiO₂...I assume they followed the approach mentioned to planarize the waveguide? A comment may be worth including if this is the case.
 - c. The text states that they deposit a 12 μ m SRN layer but the Fig. 6(a) shows this layer as being 7.5 μ m thick.

Referee #4

(Remarks to the Author)

The authors have done an excellent job addressing the reviewers' comments. The revisions have significantly strengthened the manuscript. The expanded functionalities and capabilities demonstrated here surpass prior work, but also highlight the novelty and impact of this study. I am pleased to recommend this paper for publication in its present form.

Authors' response

We would like to thank the reviewers for their thoughtful and detailed evaluations of our work. The reviewers' insightful questions and constructive suggestions have been very helpful in improving our manuscript. Below, we address each comment individually. Reviewer comments are presented *italicized*, modifications to the manuscript in **red**, and deletion from the previous version in **blue**. All substantial changes to the manuscript are listed in the response below unless otherwise noted.

Reviewer #1

This is an interesting paper in which the authors present a method to program the nonlinearity of a planar waveguide by illuminating it with different light patterns. The waveguide material is SiN, that exhibits third order nonlinearity, and the effective second order nonlinearity is achieved by external biased voltage. The illumination locally changes the conductivity of a photon-conductive layer on top of the waveguide, thereby generating a patterned bias field, and therefore a modulated effective second order nonlinearity. The ability to program the nonlinearity is demonstrated in a series of experiments, including quasi-phase-matched (QPM) second harmonic generation with different QPM periods, engineering the spectral, the spatial or the spatio-spectral properties of the generated light, feedback loop to track variations of the pump wavelength and more.

Overall, I believe that this is a significant new result, with important potential applications. QPM processes are widely used nowadays in many applications, but to large extent, the properties of the nonlinear crystal are fixed at the fabrication process and cannot be altered. This paper presents a practical method to obtain a programmable nonlinear crystal. The parameters of the crystal (effective nonlinearity of 0.47 pm/V) are still about an order of magnitude lower than those of periodically poled ferroelectrics, but the authors discuss in the supplementary methods to further increase it.

I therefore believe that this paper can be considered for publication in Nature after properly addressing the following points:

Reply 1.0: We thank the reviewer for the recognition of our contributions and an accurate summary of the work.

Reviewer Comment 1.1 — *Most of the experimental results are given in arbitrary units. What was the pump power in mW? and what was the measured second harmonic power in mW? This info should be provided.*

Reply 1.1: We thank the reviewer for the helpful comment. Using the measurement data of the total power, we now present the intensity of the generated second harmonics in physical units. The corresponding figures have been modified to reflect this change. Furthermore, we provide detailed information about the pump laser, including its power, repetition rate, and pulse duration, in the captions of the figures. The changes made can be seen in the boxes below.

Box 1: New text in the caption of Fig. 2

For all the measurements, the bias voltage was 1000 V. The nominal on-chip pump power, inferred from the transmission, ranged between 2.2 mW and 4.5 mW depending on the wavelength, primarily due to the wavelength-dependent loss of the core.

Box 2: Change of Fig. 3

Spectral engineering of second-harmonic generation (SHG). (a) Output second-harmonic (SH) spectrum of broadband SHG pumped by ultrashort pump pulses for various illumination patterns. **The bias voltage was 500 V.** (a-i) Periodic grating with a period $\Lambda = 16.64 \mu\text{m}$, (a-ii) superposition of four monotonic grating patterns with different periods, and (a-iii) an adiabatically chirped grating pattern. Due to the rapid spatial oscillations of these quasi-phase-matching (QPM) gratings, displaying the raw illumination patterns is not visually informative. Instead, in the green patterns shown above the results plots, we present the projected QPM grating patterns downsampled to a spatial period of $17 \mu\text{m}$ in the longitudinal direction. The same applies to the patterns shown in (b). See Appendix S9 for the original (non-downsampled) illumination patterns. (b) By constructing a feedback loop based on the measured SH spectrum, we optimized the illumination pattern to obtain various target SH output spectra. Dashed lines represent the target spectrum. **The bias voltage was 800 V.** (c) The illumination pattern was updated in real-time to output a sequence of SH spectra, using pre-recorded illumination patterns. We show the results for drawing “CORNELL” in the SH spectrum, with time as the horizontal axis of the image.

The bias voltage was 800 V. See Appendix S9 for experimental details. For all the measurements, we used a pulse laser with 60 fs pulse duration and 100 MHz repetition rate. The average on-chip pump power inferred from the transmission was 6 mW.

Box 3: Change of Fig. 4

Spatial engineering of second-harmonic generation (SHG). (a) Experimental setup. The waveguide was pumped with a pulse pump laser with fixed Gaussian spatial profile, and the spatial distribution of the generated SH on a camera was measured. (b) A part of the programming

illumination pattern (left column), simulated SHG dynamics within the waveguide (middle column), and a comparison between the experimentally measured and simulated SH spatial profiles. (b-i) Monotonic grating pattern. (b-ii) Quadratically chirped grating pattern. (b-iii) Superposition of nine quadratically chirped grating patterns with transverse offsets. (b-iv) Cubically chirped grating pattern. For all the measurements, we used a pulse laser with 60 fs pulse duration and 100 MHz repetition rate. The average on-chip pump power inferred from the transmission was 6 mW. The bias voltage was 600 V. See Appendix S10 for experimental details.

Box 4: Change of Fig. 5

Spatio-spectral engineering of second-harmonic generation (SHG). (a) Experimental setup. We combined a reflective grating with a $4f$ imaging setup to record spectrally-resolved one-dimensional spatial profiles of the output second-harmonic (SH) light. The waveguide was pumped by pulses with a fixed Gaussian spatial profile. (b) Results for an illumination pattern designed to generate various numbers of spatial peaks at five different wavelengths. (c) Results for an illumination pattern designed to generate oppositely chirped Airy beams at two different wavelengths. In each of (b) and (c), the left inset shows a part of the projected grating pattern. The bottom-right inset shows the spatial distribution of the SH light at various wavelengths, marked with dashed lines in the top-right inset. **For all the measurements, we used a pulse laser with 60 fs pulse duration and 100 MHz repetition rate. The average on-chip pump power inferred from the transmission was 40 mW. The bias voltage was 600 V.** See Appendix S11 for experimental details.

Reviewer Comment 1.2 — *Many of the reported measurements cover the range of 1550-1570 nm, but only at figure S.10 of the supplementary, they report huge losses in this range. This point should be made much clearer in the paper itself. In fact, it means that the effective wavelength range of this device is quite limited at this point.*

Reply 1.2: We agree with the reviewer that optical absorption (loss) is a current limitation of our device, and we have now added material—both to the main text and to the supplementary information—to directly address this.

As shown in a supplementary figure S10, PECVD SiN, the core material of our waveguide, exhibits strong absorption with the peak value of ~ 5 dB/cm around the wavelength of 1520 nm. This is due to the characteristic absorption caused by the N-H bonds abundant in PECVD SiN [1]. Away from this absorption peak, the loss decreases to a more reasonable value of ~ 1 dB/cm for > 1550 nm. Such loss can impose limitations on the operable bandwidth of the device.

This absorption could, in principle, be alleviated by several methods, including the use of low-pressure chemical vapor deposition (LPCVD) [2], PECVD with deuterated silane [3], or high-temperature annealing [4]. Specifically, to verify the possibility of reduction of the loss, we conducted a follow-up experiment involving furnace annealing of SiN at 1200°C , where we observed that the loss decreased to 0.7 dB/cm at 1520 nm and to 0.4 dB/cm for > 1550 nm. These results show that there are approaches to circumvent the issues of optical absorption.

To address the reviewer’s comment, we have added new text to the main manuscript clarifying the limitations of the current device. Furthermore, we include measurement results of the furnace-annealed PECVD SiN as a reference for how low the absorption can be.

Box 5: New text in Sec. VII

Second, we observed relatively large optical loss, ranging from a nominal value of ~ 1 dB/cm for wavelengths > 1550 nm to a peak value of ~ 5 dB/cm around 1520 nm, due to residual hydrogen-nitrogen bonds in PECVD-deposited SiN. This limits the utility of the device for wavelengths near 1520 nm. High-temperature annealing is known to significantly reduce such absorption by driving out hydrogen impurities [4]. Furthermore, optical loss as low as 0.4 dB/m has been demonstrated in SiN nanophotonic devices fabricated with low-pressure chemical vapor deposition [2].

Prospects for reducing the loss

There are various routes to reduce the absorption of SiN from the levels observed in Sec. S4 C1. In principle, SiN is capable of achieving extremely low optical loss. The material absorption limit of SiN has been estimated to be 0.13 dB/m, and losses as low as 0.4 dB/m have been demonstrated in high-confinement SiN waveguides [2,5]. These films were fabricated using low-pressure chemical vapor deposition (LPCVD). For PECVD SiN, it is known that residual hydrogen atoms in the film increase absorption, especially near a wavelength of 1520 nm [1,6]. Reduction of loss in PECVD SiN is possible either by using deuterated silane [3] or by high-temperature furnace annealing [4], both of which reduce the hydrogen concentration in the film. Here, we present preliminary experimental results using the latter approach.

For the annealing characterization, we fabricated channel waveguides using PECVD SiN. The fabrication procedure of the waveguide mostly followed the process we used for the programmable channel waveguide (see Sec. VIII B) with two differences; We stopped the process after the first plasma etching, and we replaced the rapid thermal annealing step with a 3 hours furnace anneal at 1200 °C with N₂ flow. The final die contained two types of waveguides: straight waveguides with a length of 2 cm and a spiral waveguide with a total length of 6 cm. In Fig. S15, we show a microscope image of the spiral waveguide.

Fig. S15: A microscope image of the SiN spiral waveguide. The image was created by stitching together nine individual images with smaller fields of view to capture the entire waveguide.

For the measurement of optical loss, we coupled a CW laser with a tunable wavelength between 1500 nm and 1630 nm into the waveguides and measured the power at the output. By dividing the ratio of output power by the difference in the lengths of the waveguides, we estimated the propagation loss per unit length. The measurement results are shown in Fig. S16. Though the data came out slightly noisy, potentially due to the multi-mode nature of the waveguide, we observed a clear reduction in optical loss—down to 0.4 dB/cm for wavelengths > 1550 nm and to 0.8 dB/cm around 1520 nm, where we had previously observed losses of 5 dB/cm. In the future, these loss values may be further improved by optimizing the etching recipe to reduce contributions from sidewall roughness.

Fig. S16: Optical loss of PECVD SiN waveguides after furnace annealing, estimated by comparing the power transmission through two waveguides of different lengths. We used a straight channel waveguide with a length of 2 cm and a spiral waveguide with a length of 6 cm.

Reviewer Comment 1.3 — *The authors present in Section IV an iterative method to shape the spectrum of the generated light. It should be noted that this is not essential, since there is an analytical method to design the phase matching spectrum, see for example Shiloh et al, Optics Letters 37, 3591 (2012); Leshem et al, Optics Letters 39, 5370 (2014).*

Reply 1.3: We thank the reviewer for bringing important references to our attention. The suggested references [7, 8] demonstrate that one can analytically construct QPM grating structures to generate desired spectral-temporal waveforms in a systematic manner. For the types of tasks we demonstrate in the main text, analytic designs of QPM gratings could suffice, provided that one has good characterization of the system, e.g., the shape of the pump pulse, waveguide dispersion, and fabrication tolerances. On the other hand, a main advantage of our iterative in situ optimization approach is its ability to compensate for potential experimental imperfections and miscalibrations. We have updated the text in Sec. 4 to incorporate this context.

Box 7: Change in Sec. IV

For more general spectral features, Refs. [7, 8] presented and demonstrated frameworks for analytically designing QPM grating structures. These frameworks could be employed to design patterns for the programmable device we present.

Up to this point, the illumination patterns we used to program the waveguide were designed manually, in that we designed them based on standard knowledge of NLO and what $\chi^{(2)}(x, z)$ patterns would yield the desired SHG processes. To demonstrate the ability to shape the output SHG spectrum in ways that are likely impractical using conventional NLO devices, which don't support real-time reconfiguration, we arbitrarily shaped and dynamically updated the SHG spectrum. We achieved this by constructing a real-time feedback loop between the broadband SH spectral measurement and update of the programming illumination patterns (Fig. 3(b)) (see Appendix S9 for details on the optimization procedure used for this in situ inverse design). **This approach enables**

spectral engineering in a way that is robust against imperfections and miscalibrations in the device and experimental setup.

Reviewer Comment 1.4 — *In the supplementary section, Fig. S3, the authors suggest a method to improve the performance using a photoconductive core. Wouldn't such a device suffer from very high losses? Moreover, these losses will be dependent on the QPM pattern.*

Reply 1.4: We thank the reviewer for the comment. We agree that it is possible that carriers induced by the programming illumination may cause increased optical loss in the waveguide, and this is an important caveat for this potential future device. This potential approach based on a photoconductive core is still preliminary, and we expect that further material investigations will be necessary to unravel the tradeoff space between photoconductivity and optical loss. In the new text added to the manuscript (shown in Box 8), we discuss these limitations and potentially relevant factors.

Box 8: New text in Sec. S3

A possible drawback of this approach is increased optical loss. As the programming illumination excites free carriers inside the core, it would induce free-carrier absorption (FCA) in a manner that depends on the illumination pattern. It is possible that such FCA imposes a limit on how low the loss can be with this approach. Although this approach is still preliminary and requires further research, we discuss several factors that we believe are notable. First, we note that the level of photoconductivity required for programmable photonics is quite low. As shown in Fig. S5, only a 10× contrast between the bright and dark states is needed to achieve sufficient programmability. This relaxed requirement opens up the possibility of employing materials with lower carrier concentrations, extending beyond the conventional catalog of photoconductive materials. Indeed, SRN, which we used in our work, is typically not employed in conventional photoconductor applications due to its low photoconductivity. Second, as seen in indium tin oxide, the presence of free carriers does not necessarily preclude optical transparency—this is made possible by its large bandgap and high plasma frequency. This underscores the importance of co-engineering optical and electrical properties for the development of an ideal core material.

Reviewer Comment 1.5 — *Can this method be used with a channel waveguide, instead of a planar waveguide? This can potentially provide much higher conversion efficiency.*

Reply 1.5: We thank the reviewer for this question/suggestion. In this work, we employed a planar waveguide geometry to demonstrate the full flexibility of programmable nonlinear photonics, accessing both spatial and spectral features of light. However, this approach suffers from low conversion efficiency. For applications that do not require spatial engineering, as pointed out by the reviewer, it is highly advantageous to use a channel waveguide structure.

We have now conducted a follow-up experiment to assess the practicality and promise of applying our programming approach to channel waveguides. We fabricated a programmable nonlinear *channel* waveguide and, by pumping the device with a tunable-wavelength CW laser and measuring the generated SHG, we clearly observed programmable phase matching of second-harmonic generation with a conversion efficiency at least 40 times higher than that of the planar waveguide. We believe these ex-

perimental results answer the reviewer’s question positively, demonstrating that our method can indeed be applied to channel waveguides to achieve higher conversion efficiency.

Below, we indicate how the manuscript has been revised to incorporate these new results.

Box 9: New text in the “Methods” section

Fabrication and characterizations of programmable nonlinear channel waveguides

In the main text, we primarily focused on a programmable planar waveguide. Although this design allowed us to engineer spatial features of light, it came with the drawback of lower light confinement, which reduced the conversion efficiency. One possible solution to this challenge is to fabricate a channel waveguide structure to prevent spatial diffraction, which enables light propagation over longer distances while maintaining tight transverse confinement. To demonstrate the compatibility of our approach with this channel waveguide geometry, we fabricated a prototypical programmable channel waveguide using SiN and characterized its performance.

The device was fabricated with the following steps. First, Silicon Valley Microelectronics provided a Si substrate with a 1 μm bottom oxide cladding and a 2 μm PECVD-grown SiN layer, identical to the substrate type used for the programmable planar waveguides. Next, we deposited a 500 nm-thick SiO₂ layer using PECVD, followed by sputtering a 200 nm-thick Cr layer. For photolithography, we spin-coated a deep ultraviolet (DUV) anti-reflective coating and photoresist. Exposure was performed using a DUV stepper (PAS 5500; ASML), after which we developed the photoresist and wet-etched the Cr to form a hard mask. We then etched through the SiN layer using CHF₃/O₂ gases in a plasma etcher (PlasmaPro 100 RIE; Oxford Instruments). To deposit additional cladding on the SiN sidewalls and thereby reduce losses into the photoconductive layer, we deposited 3 μm of SiO₂ with PECVD, and subsequently etched away 2.5 μm of oxide using the same plasma etching recipe described above. At this stage, we performed rapid thermal annealing at 800 °C for 5 mins. Finally, we deposited a 12 μm layer of SRN and a 20 nm layer of ITO as a transparent electrode. A cross-sectional scanning electron microscope (SEM) image of the fabricated channel waveguide is shown in Fig. 7. The SiN core was approximately 4 μm wide and 2 μm high. The waveguide had an overall length of 1.5 cm, with the programmable region limited to approximately 7 mm due to constraints of the imaging setup.

Fig. 7 An SEM image of the cross section of a programmable nonlinear channel waveguide. We

highlight the boundaries between different materials with dashed lines.

We then conducted a nonlinear-optical characterization of the device. We coupled CW pump light with various wavelengths between 1540 nm and 1600 nm and measured the generated SHG while scanning the period of the monotonic QPM grating. We calculated the normalized SHG conversion efficiency by dividing the detected SHG power by the pump power squared measured after the waveguide, including appropriate corrections for propagation losses and collection inefficiencies (see Appendix S8 for details). The measurement results are shown in Fig. 8, which clearly demonstrate the capability to achieve phase matching at a desired pump wavelength by adjusting the QPM grating period.

Fig 8: Normalized SHG conversion efficiency of the programmable nonlinear channel waveguide for various pump wavelength (legend) and programmed QPM grating periods. Approximately 4 mW of pump power was detected after the waveguide. We used 1500 V of bias voltage with 10 Hz of frequency.

Note that our waveguide core measures approximately $2\ \mu\text{m} \times 4\ \mu\text{m}$ in cross section, supporting multiple spatial modes even at the pump wavelength. Moreover, the core’s skewed aspect ratio made it difficult to couple the pump light exclusively into the fundamental mode. Consequently, the pump power measured after the waveguide could include contributions from higher-order modes that do not contribute to SHG, which explains the wavelength-dependent variations in the normalized conversion efficiency. Thus, the efficiencies shown in Fig. 8 should be considered a lower bound of the device performance. The maximum normalized conversion efficiency observed was $\eta_{\text{norm}} = 2 \times 10^{-3} \text{ %/W}$, corresponding to a normalized slope conversion efficiency of $\eta_0 = 4 \times 10^{-3} \text{ %/W/cm}^2$, showing an approximately 40-fold improvement over the programmable nonlinear planar waveguide results presented in the main text. We note that reducing the core thickness below the micron scale to achieve single-mode operation causes the first-order poling period to fall below the smallest programmable feature size. If single-mode operation is prioritized, higher-order quasi-phase matching can be employed, albeit with reduced conversion efficiency. At the same time, the tighter field confinement increases nonlinear coupling and could compensate for

the reduction of conversion efficiency.

Reviewer #2

I co-reviewed this manuscript with one of the reviewers who provided the listed reports.

Reply 2.0: We thank the reviewer for providing a review of our manuscript. Please refer to the corresponding section of this document for our response to the comments.

Reviewer #3

The authors demonstrate a versatile approach to reconfigurable nonlinear frequency conversion based on the effective $\chi(2)$ nonlinearity realized through the DC-Kerr effect. Because the DC field can be electronically reconfigured in space and time, this allows the effective $\chi(2)$ nonlinearity to be reconfigured. The authors demonstrate programmable quasi-phase matching for second harmonic generation, broadband and inverse-designed SHG profiles (i.e., spectral engineering), control of the spatial profile of the SHG light (spatial engineering), and combined spatio-spectral engineering.

To realize such re-configurability of the DC electric field, the authors develop a clever approach that does not require any fine fabrication, but instead relies on a single unpatterned electrode and a photoconductive layer that, when illuminated by a structured field, results in a structured DC field that reaches the waveguide core.

Overall, I found this work to be creative and original, and the manuscript is well-written and easy to follow. The results are intuitive, pleasing, and of high quality. In contrast to the many nonlinear photonics works which optimize performance for a limited set of scenarios, here the authors maximize flexibility and re-configurability. They use an extremely simple device platform, with most of the complexity shifted to the SLM-based optical projector system. This approach doesn't really compete with nonlinear integrated photonics devices, where the point is to limit off-chip resources, but is perhaps more suited for applications in which periodically-poled crystals that are part of a free-space optical system are used.

Reply 3.0: We thank the reviewer for the recognition of our contribution and accurate summary of our work.

Reviewer Comment 3.1 — *This brings me to my main reservation for publication in Nature, which is that I don't know what problem(s) this work actually solves. The observed efficiency of $5 \times 10^{-5} \%$ /W is very low (8 orders of magnitude lower than the highest efficiencies shown with the DC-Kerr effect in SiN and around 5 orders of magnitude lower than even inefficient poled crystals), which renders it hard to use in almost any application, I believe. Supplement S1 details how the magnitude of the $\chi(2)$ nonlinearity can be improved by a factor of 50x through changes in materials, etc, but even with this, the efficiencies would appear to be very low. And the applications the authors have claimed are all described very qualitatively, without any hard metrics outlined that*

could help a reader understand if the approach described in this paper can be truly relevant if the advances in Supplement S1 are realized. I believe that the authors need to more thoroughly address which applications their approach is particularly well-suited for, the level of performance needed for that application (or at least, what is currently available), and how they can get there, for this work to rise to the level of Nature. While it is true that not all Nature papers need clear applications, given that there aren't really any new physical principles being explored and that the work is essentially a technology demonstration, I think the above is necessary.

Reply 3.1: Given the technological aspect of our work, we agree with the reviewer that it is important for us to show how our work could be used to solve outstanding problems. The reviewer's comment has given us an opportunity to reflect and incorporate a list of unique applications backed by quantitative analysis, which we believe has greatly strengthened the manuscript.

Summary of reply: We first put in context the nonlinearity and conversion efficiency we have achieved, comparing it to other state-of-the-art works using the DC-Kerr effect in SiN. In short, our reported induced $\chi^{(2)}$ value is higher than almost all reported values in the literature for DC-Kerr effect devices in SiN, and within 10% of the record highest value reported. Our conversion efficiency in a planar waveguide is low compared to demonstrations in channel waveguides or in microring resonators because of the low field confinement in planar waveguides. However, in our revised manuscript, we now also report a programmable nonlinear channel waveguide and show conversion efficiency per length greater than the landmark result from Bres's group at EPFL [9]. We then propose four applications of our approach to programmable nonlinear photonics and give detailed calculations showing how our platform, even without assuming increased nonlinearity, could achieve state-of-the-art performance numbers in some regimes of practical interest, while simultaneously being programmable—which for some applications is either a necessity or at least very convenient.

For applications of nonlinear photonics, nonlinearity is of course an important figure of merit that determines what functions a platform can meaningfully perform. In this context, we would like to note that the programmable $\chi^{(2)}$ nonlinearity demonstrated in our work is already reasonably high among the reported demonstrations based on the DC-Kerr effect (i.e., electric-field-induced $\chi^{(2)}$ nonlinearity). For instance, Ref. [10] reports an SHG efficiency of $\eta_{\text{norm}} = 2500\%/W$, nearly 8 orders of magnitude greater than our reported efficiency. To our knowledge, this is the highest SHG efficiency achieved using the DC-Kerr effect in SiN. However, the induced $\chi^{(2)}$ nonlinearity reported in Ref. [10] was $\chi^{(2)} = 0.2\text{ pm/V}$. In comparison, our work demonstrated a programmable $\chi^{(2)} = 0.47\text{ pm/V}$.

Why then is there such a huge gap in the conversion efficiency when the nonlinear coefficients are similar? This is because conversion efficiency is a quantity that depends very strongly on the device geometry, sometimes even more than nonlinearity. To see this more clearly, we summarize notable demonstrations of the DC-Kerr effect with SiN nanophotonics in Table R1. For instance, a channel waveguide geometry can enhance the transverse confinement of the light, increasing the conversion efficiency compared to a bulk crystal or a planar waveguide. Using an SiN channel waveguide, Ref. [9] demonstrated $\eta_{\text{norm}} = 5 \times 10^{-2}\%/W$. To demonstrate that our approach is compatible with such geometry, we fabricated a programmable channel waveguide during the revision of the manuscript (see Box 11), where we observed $\eta_{\text{norm}} = 2 \times 10^{-3}\%/W$, which is a 40 times improvement from our demonstration with a planar waveguide. Here, our conversion efficiency is still lower by an order of magnitude than what is reported in Ref. [9], but this is due to the large length of their device totaling

Device type	Induced $\chi^{(2)}$ non-linearity	Conversion efficiency η_{norm}	Reference
Programmable planar waveguide	0.47 pm/V	$5 \times 10^{-5} \%$ /W	This work (Fig. 2b)
Programmable channel waveguide	0.47 pm/V	$2 \times 10^{-3} \%$ /W	This work (Fig. 8)
Channel waveguide	0.5 pm/V	$5 \times 10^{-3} \%$ /W	Ref. [11]
Channel waveguide	0.3 pm/V	$5 \times 10^{-2} \%$ /W	Ref. [9]
Microring resonator	0.03 pm/V	47.6 %/W	Ref. [12]
Microring resonator	0.2 pm/V	2500 %/W	Ref. [10]
Microring resonator	Not provided	651 %/W	Ref. [13]
Microring resonator	0.03 pm/V	3.2 %/W	Ref. [14]
Microring resonator	0.022 pm/V	141 %/W	Ref. [15]

Table R1: Summary of the performance of electric-field induced $\chi^{(2)}$ nonlinear-optical device on SiN. In all references but our work, $\chi^{(2)}$ nonlinearity stemmed from the DC-Kerr effect induced by the photogalvanic effect.

5.8 cm (while ours is 0.7 cm). In terms of the slope conversion efficiency (i.e., conversion efficiency normalized by the device length squared), our device showed $\eta_0 = 4 \times 10^{-3} \%$ /W/cm², which is greater than the $\eta_0 = 3 \times 10^{-3} \%$ /W/cm² reported in Ref. [9].

The efficiency could be further boosted by forming resonator structures by a curved waveguide, which recirculates the light. This simultaneously increases the pump field intensity and effective interaction length, resulting in a steep scaling of $\eta_{\text{norm}} \propto Q_{\text{FH}}^2 Q_{\text{SH}}$ (Q_{FH} and Q_{SH} are Q-factors of the fundamental harmonic (FH) and second harmonic (SH), respectively). This would enable another nearly 5 orders of magnitudes increase of conversion efficiency compared to a channel waveguide, as shown in the table.

However, this does not mean that devices without resonant structures are less “useful” than microring resonators. For example, frequency conversion of ultrashort pulses with micro-ring resonators require careful synchronization of pulse repetition rate with the cavity free spectral range and is not suitable for pulse-by-pulse operation. On the other hand, single-pass waveguides offer a much simpler solution, where their apparently low conversion efficiency can be compensated by the high peak powers available under pulsed excitation. A similar trade-off applies when comparing devices with low transverse confinement—such as free-space nonlinear optics or planar waveguides—to highly confined channel waveguides. While the former exhibit lower η_{norm} , their large mode volumes allow them to handle significantly higher pump powers, which can compensate for the lower efficiency and are more suitable for applications that require higher optical powers. For quantum light generations, spatial degrees of freedom in free-space nonlinear optics or planar waveguides allow convenient means to separate entangled photon pairs into different paths [16, 17]. It is true that one usually prefers a device with higher conversion efficiency over one with lower efficiency when everything else is held equal, but this is not the only consideration for applications. Different applications can benefit from different device types, and have different standards and requirements for performance. However, it is not trivial to evaluate and compare the utility of different device platforms for different applications, and in the end careful quantitative evaluation is needed to be able to assess if, when, and how a device platform will be useful for a particular application. The reviewer’s comment has helped us recognize that our initial manuscript fell short in this respect.

We have now performed several quantitative evaluations using back-of-the-envelope calculations, and it turns out that we could, in principle, achieve competitive and sometimes even state-of-the-art

performance in various applications **while conservatively assuming only the nonlinearity and the slope conversion efficiencies already experimentally demonstrated in this work**. Some of the positive evaluations are due to the potential of SiN to realize ultra-low-loss waveguides, which allows the fabrication of ultra-high-Q resonators or meter-long waveguides on chip [2, 18, 19]. Such extended interaction lengths can compensate for the nonlinearity being weaker compared to more established nonlinear materials, while simultaneously increasing the number of programmable parameters. The list below summarizes four potential applications that we believe are suitable for programmable nonlinear photonics using our techniques.

- **Potential Application 1: Highly programmable on-chip optical pulse shaper**

Using a long programmable channel waveguide with engineered QPM gratings, we could generate optical pulses with arbitrary spectral-temporal profiles using SHG on a fully integrated platform. We estimate that this on-chip optical pulse shaper could realize a programmable temporal window size of 300 ps with a minimum feature size of 200 fs. This amounts to 1500 programmable parameters, which is larger than many state-of-the-art demonstrations in the field. The energy efficiency of the device can be made near unity with moderate pump power.

- **Potential Application 2: Programmable quantum frequency converter**

With a long programmable channel waveguide, we could realize a quantum frequency converter (QFC), which transduces the frequency of a quantum optical signal. The ability to reconfigure the phase-matching condition allows flexible routing of the quantum signal to different frequency bands, enabling dense wavelength-division multiplexing [20]. The estimated conversion efficiency exceeds that of a commercial, non-programmable QFC, and we expect unit-efficiency conversion to be possible with a moderate 0.7 W of quasi-CW pump.

- **Potential Application 3: Widely tunable and highly efficient integrated light sources**

With a resonator composed of a programmable waveguide, we could realize resonant-enhanced frequency conversion with widely reconfigurable phase-matching conditions. When state-of-the-art loss values for SiN are assumed, the device is expected to exhibit a normalized SHG conversion efficiency of $\eta_{\text{norm}} = 1.3 \times 10^7 \%$ /W and a threshold for optical parametric oscillation of $P_{\text{th}} = 7.7 \mu\text{W}$. These numbers are better than the record values demonstrated in integrated photonics.

- **Potential Application 4: Programmable quantum light sources**

We can realize quantum light sources with engineerable photon-photon correlation structures via the spontaneous parametric downconversion process (SPDC) with programmable QPM gratings. Quantitative estimates show that the levels of brightness necessary for realistic applications can be achieved with a moderate amount of pump power.

Below, we provide full details on the quantitative analysis of each potential application.

Potential Application 1: Highly programmable on-chip optical pulse shaper

Building an optical pulse shaper that can access both a large programmable time scale T_{window} and

temporal resolution ΔT is a challenging task because the device needs to cover physical processes with vastly different timescales. The ratio between these timescales, known as the time-bandwidth product $BT = \Delta T/T_{\text{window}}$, is a figure of merit that corresponds to the number of programmable parameters of a pulse shaper [21]. In Table R2, we summarize the performance of notable demonstrations of optical pulse shapers. Here, we show that a programmable channel waveguide could potentially realize a competitive on-chip pulse shaper with as many as 1500 programmable parameters, whose expected performance is shown in the table. Our proposed approach offers sub-ps timing resolution (i.e., terahertz of bandwidth) and hundreds picoseconds of programmable time window (i.e., gigahertz-level frequency resolution), which could be heterogeneously integrated in a larger photonic system as flexible on-demand on-chip pump source. Beyond applications in photonics, optical pulse shaping in such a frequency range has been used for high-frequency radio-frequency (RF) and terahertz wave generation. That is, by detecting the synthesized optical pulses with high-bandwidth photodetectors, the photocurrent inherits the temporal-domain intensity profile of the optical pulse [22, 23]. The accessible bandwidth of the generated electronic signal can go far beyond what is possible through purely electronic means.

Device type	Time window T_{window}	Time resolution ΔT	Programmable parameters (BT)	Reference
Programmable channel waveguide (theory)	300 ps	200 fs	1500	This work
Commercial Fourier transform optical pulse shaper	3.2 ps	5 fs ^a	640	Ref. [24]
Meta-surface Fourier transform optical pulse shaper	6.6 ps	10 fs ^b	660	Ref. [25]
Commercial acousto-optic programmable dispersive filter	14 ps	5.3 fs ^c	2600	Ref. [26]
Free-space line-by-line pulse shaper	380 ps ^d	2 ps	190	Ref. [23]
Integrated line-by-line pulse shaper	40 ps ^e	1.25 ps	32	Ref. [27]

Table R2: Summary of the performance of optical pulse shapers. ^aCalculated from the maximum spectral window of 600 nm around 1000 nm. ^bCalculated from the spectral window between 700 nm and 900 nm. ^cCalculated from the spectral window between 650 nm and 1100 nm. ^d2.6 GHz spectral resolution. ^e25 GHz channel spacing.

A possible integration of a pulse shaper based on a programmable waveguide is shown in Fig. R1. High-power pump pulses are coupled into the SiN programmable channel waveguide. The programmable QPM grating then engineers the shape of the SHG pulse into desired forms [7, 8, 28]. With programmable illumination implemented using micro-LED arrays [29] and pump light source provided by micro-combs, the entire pulse shaper setup could be fully integrated on-chip. To maximize the programmability of the device, we consider maximizing its length. The superb properties of SiN allow access to 0.4 dB/m loss [2], and spiral-shaped waveguides with more than a meter of length have been demonstrated on-chip [18, 19]. Here, we assume $L_{\text{tot}} = 1.5$ m of spiral-shaped waveguide with dispersion engineering to minimize group-velocity dispersion. While performing periodic poling on such a long and winding structure would be challenging with conventional means, the real-time programmability of our approach

allows robust optimization of QPM gratings based on experimental feedback, as demonstrated in our work. Assuming a moderate group-velocity mismatch of $GVM = -0.2 \text{ ps/mm}$ and using the pulse shaping scheme with QPM grating [28], the device could engineer arbitrary SH pulse shapes with a temporal window as large as $T_{\text{window}} = L_{\text{tot}} \times GVM = 300 \text{ ps}$. The smallest programmable temporal feature size is set by the duration of the pump pulse, $\Delta T = T_{\text{pump}}$. Assuming $T_{\text{pump}} = 200 \text{ fs}$, the time-bandwidth product (BT) of the pulse shaper (i.e., the number of programmable spectral-temporal features) would become $BT = T_{\text{window}}/\Delta T = 1500$. These numbers are summarized in Table R2.

Figure R1: An illustration for the possible future implementation of on-chip pulse shaper with a long programmable channel waveguide.

Finally, we comment on the efficiency of our proposed pulse shaper. To verify that a channel waveguide can enhance the conversion efficiency, we newly fabricated a SiN programmable channel waveguide during the revision of the manuscript (see Box 11 below). On this waveguide, we observed a normalized SHG conversion efficiency of $\eta_{\text{norm}} = 2 \times 10^{-3} \text{ \%}/\text{W}$, or equivalently a slope conversion efficiency of $\eta_0 = 4 \times 10^{-3} \text{ \%}/\text{W}/\text{cm}^2$, showing a 40-fold improvement over the programmable planar waveguide. We assume that the proposed pulse shaper consists of a channel waveguide with the same nonlinearities as demonstrated in our work.

Because there is a finite group-velocity mismatch, we need to account for the effects of temporal walk-off in the calculation of conversion efficiency. More specifically, since the FH and SH pulses temporally walk off every $L_{\text{walk-off}} = T_{\text{pump}}/|GVM| = 1 \text{ mm}$ of propagation, the peak SH power will be approximately $P_{\text{SH}} = \eta_0 P_{\text{FH}}^2 L_{\text{walk-off}}^2$, where P_{FH} is the peak power of the FH mode. The total energy conversion efficiency of the device, accounting for temporal walk-off (but assuming no pump depletion), is approximately given as

$$\eta_{\text{tot}} = \frac{P_{\text{SH}} T_{\text{window}}}{P_{\text{FH}} T_{\text{pump}}} = \frac{\eta_0 P_{\text{FH}} T_{\text{pump}} L_{\text{tot}}}{|GVM|}, \quad (\text{R1})$$

which reaches a conversion efficiency of $\eta_{\text{tot}} = 10\%$ with a peak power of $P_{\text{FH}} = 170\text{ W}$. Ultrashort pulses with such peak powers are straightforward to obtain from fiber-based mode-locked lasers (for instance, the ELMO HP laser we used in our work outputs more than 30 kW of peak power), indicating that the entire programmable arbitrary optical-pulse synthesizer system could be integrated into a compact setup with no free-space optical components. Also, with a relatively large cross-section of $2\ \mu\text{m} \times 4\ \mu\text{m}$, this waveguide can handle 60 kW of peak power when pumped by femtosecond pulses [30].

Potential Application 2: Programmable quantum frequency converter

A quantum frequency converter (QFC) is a device that converts the frequency of a photon's quantum-optical signal [31]. One of the major applications of QFCs is quantum networking, where the frequency of photons emitted from quantum nodes—often composed of atoms or vacancy centers—is converted to a frequency more suitable for long-range communication, and vice versa [32]. For this application, the ability to tune the target wavelengths is an essential feature of a QFC. It not only allows compensation for the inherent inhomogeneities of the transition frequencies of certain quantum emitters but also enables wavelength division multiplexing (WDM) to boost communication capacity. Recently, Cisco Quantum Labs published a roadmap for scalable quantum networking with dense WDM [20], where reconfigurable quantum interfaces—i.e., QFCs with tunable phase-matching conditions—play a central role. However, a suitable device has not yet been demonstrated. Our work could enable such an application, thanks to the fast and extremely wide programmability of the phase-matching conditions. More broadly, Ref. [33] reviews the unique role that nontrivial QPM structures can play in QFC applications, many of which can be implemented and dynamically switched on our platform. Below, we quantitatively show that a SiN programmable channel waveguide could realize high-efficiency conversion of quantum signals using sum/difference frequency (SFG/DFG) processes.

As illustrated in Fig. R2, we consider the same device design as discussed above for “Potential Application 1: Highly programmable on-chip optical pulse shaper,” but operated differently. The device consists of a programmable SiN channel waveguide with a slope SHG conversion efficiency of $\eta_0 = 4 \times 10^{-3}\ \%/W/\text{cm}^2$ (as demonstrated in this work; see Box. 11) and a total length of 1.5 m. We consider three different modes of light labeled as the “atomic mode”, “communication mode”, and “pump mode”, with frequencies ω_a , ω_c , and ω_p , respectively. The goal of a QFC is to convert photons in the atomic mode to the communication mode via a DFG process $\omega_a - \omega_p = \omega_c$. Note that this process also transduces photons in the communication mode to the atomic mode at the same time. Assigning annihilation operators to the atomic and communication modes as \hat{a} and \hat{c} , the DFG process induces the following transformation:

$$\hat{c}(L) = \cos(\sqrt{\eta_{0,\text{DFG}}P_pL})\hat{c}(0) + \sin(\sqrt{\eta_{0,\text{DFG}}P_pL})\hat{a}(0), \quad (\text{R2})$$

where $L = 1.5\text{ m}$ is the length of the waveguide, P_p is the power of the pump field, and $\eta_{0,\text{DFG}}$ is the DFG slope conversion efficiency. Assuming weak wavelength dependence of the nonlinearity, $\eta_{0,\text{DFG}} = 4\eta_0$ holds, with the slope SHG conversion efficiency of the waveguide $\eta_0 = 4 \times 10^{-3}\ \%/W/\text{cm}^2$. Full frequency conversion is achieved when $\hat{c}(L) = \hat{a}(0)$, corresponding to $P_p = 0.7\text{ W}$. Such a power level can be straightforwardly achieved using quasi-CW pulsed operation. The normalized conversion efficiency of the device is $\eta_{0,\text{DFG}} \times (1.5\text{ m})^2 = 360\ \%/W$, which exceeds the value $80\ \%/W$ of a commercially available QFC [34]. For a group-velocity mismatch of -0.2 ps/mm , the phase-matching bandwidth is expected to be 3.2 GHz. Such an extremely narrow bandwidth, enabled by the long waveguide, can play the role of a filter, rejecting undesired background photons [35]. Overall, we expect the realization of such a programmable QFC to be feasible on SiN. The conversion efficiency η_0 we assumed was observed

in our work. Furthermore, SiN waveguides longer than a meter have been demonstrated in various studies [18,19], and optical loss as low as 0.4 dB/m has been achieved [2]. An ability to reconfigure the QPM grating based on real-time experimental feedback would facilitate efficient quasi-phase matching on such a long waveguide.

Figure R2: Illustration of a possible implementation of a QFC with a programmable nonlinear waveguide. The pump light drives a frequency conversion of a weak input quantum signal to the communication mode.

Potential Application 3: Widely tunable and highly efficient integrated light sources

Nonlinear $\chi^{(2)}$ nanophotonic resonators hold the key to highly efficient integrated light sources. For second-harmonic generation, the highest normalized conversion efficiency to date has reached $5 \times 10^6 \text{ \%}/\text{W}$, which was demonstrated in thin-film lithium niobate [36]. The performance of other nonlinear optical processes, such as optical parametric oscillation [37], can also be significantly improved by high-Q resonators. A crucial drawback of such integrated photonics is the lack of tunability. That is, the phase-matching condition is typically fixed as the QPM grating period at the stage of device fabrication, making each device useful only for a narrow wavelength bandwidth. Recently, all-optical poling (AOP), induced by the combination of DC Kerr and photogalvanic effects, has attracted attention as a solution to this challenge [11]. By leveraging AOP, Ref. [15] demonstrated efficient green-light generation over a 2.6 THz tunable bandwidth. At the same time, the AOP-based approach still faces several challenges. First, AOP involves a complicated buildup of interference patterns, which takes several seconds to reconfigure [15]. Second, the formation of AOP requires efficient SHG, but one may be interested in other nonlinear-optical processes—e.g., spontaneous parametric downconversion (SPDC). In Ref. [13], the authors pumped the resonator with FH light to produce AOP, then switch the pump wavelength to SH to realize SPDC. Furthermore, they observed that the effective $\chi^{(2)}$ nonlinearity of the AOP faded over time scales of $40 \text{ s} \sim 70 \text{ s}$ during SPDC operation. The rate of decay increased with stronger pump fields, and they hypothesized that the effect was caused by multiphoton excitation from the pump. This

could hinder efforts to increase pump power to achieve stronger gain—for example, in the generation of squeezed states or for realization of optical parametric oscillators.

Our approach based on programmable $\chi^{(2)}$ nonlinearity can provide a resolution to these challenges. While benefiting from the superb optical properties of the SiN photonics platform, we could induce arbitrary QPM grating patterns with an update speed potentially approaching 200 Hz. Furthermore, because the QPM grating is produced by an external bias electric field, there is no issue with fading of the QPM grating even when no SHG is taking place, making it attractive for applications like SPDC. Below, we show that a microring resonator made of a SiN programmable nonlinear waveguide can indeed achieve competitive performance for various nonlinear optical processes, enabling widely tunable and highly efficient integrated light sources.

In Fig. R3, we depict how such a device may be integrated. For concreteness, we consider a microring resonator with $R = 100 \mu\text{m}$ radius, composed of a curved channel waveguide structure. Though the loss of the programmable planar waveguide was rather high ($> 1 \text{ dB/cm}$), we verified in a follow-up experiment shown in Box 12 below that high-temperature annealing of the film allows us to reduce the loss further to 0.4 dB/cm . This corresponds to a field decay rate of $\kappa_{\text{FH}} = 2\pi \times 109 \text{ MHz}$, assuming a group index of 2. The intrinsic Q-factor of the resonator for the FH is thus $Q_{\text{FH},0} = \frac{\omega_{\text{FH}}}{2\kappa_{\text{FH}}} = 8.9 \times 10^5$. Assuming the same Q-factor values for the SH and critical coupling for both modes, the loaded Q-factors of the resonator are expected to be $Q_{\text{FH}} = Q_{\text{SH}} = 4.4 \times 10^5$. The normalized SHG conversion efficiency of the device takes the form

$$\eta_{\text{norm}} = \frac{8g^2 Q_{\text{FH}}^2 Q_{\text{SH}}}{\hbar\omega_{\text{FH}}^4}, \quad (\text{R3})$$

where g denotes the nonlinear coupling constant [36]. The value of g can be calculated via

$$g = \sqrt{\frac{\hbar\omega_{\text{FH}}\eta_0 v_g^3}{4\pi R}}, \quad (\text{R4})$$

where η_0 is the normalized slope conversion efficiency of the waveguide [38]. For the programmable channel waveguide, we demonstrated $\eta_0 = 4 \times 10^{-3} \text{ \%}/\text{W}/\text{cm}^2$, leading to $g = 2\pi \times 1.9 \text{ kHz}$. Plugging in these numbers, we find $\eta_{\text{norm}} = 42 \text{ \%}/\text{W}$ using only values demonstrated in this work. It is worth noting that SiN is known for its superb ability to achieve extremely low-loss photonic devices. SiN microring resonators with intrinsic Q-factors exceeding 60 million have been reported [5]. Considering the steep scaling of $\eta_{\text{norm}} \sim Q^3$, this suggests the potential to reach $\eta_{\text{norm}} = 1.3 \times 10^7 \text{ \%}/\text{W}$ with loaded Q-factors of 30 million. Notably, this value exceeds the highest recorded SHG conversion efficiency of $5 \times 10^6 \text{ \%}/\text{W}$ on integrated photonics, which was demonstrated on a thin-film lithium niobate microring resonator [36].

Another important application of $\chi^{(2)}$ nonlinear photonics is the optical parametric oscillator (OPO), where short-wavelength pump light parametrically downconverts to produce signal light at a longer wavelength. An important figure of merit for an OPO is the threshold power P_{th} , above which macroscopic signal light is generated. The lowest OPO threshold reported to date on an integrated platform is $P_{\text{th}} = 30 \mu\text{W}$, demonstrated in Ref. [37] on a thin-film lithium niobate microring resonator. Assuming critical coupling, the OPO threshold takes the form

$$P_{\text{th}} = \frac{\hbar\omega_{\text{FH}}^4}{8g^2 Q_{\text{FH}}^2 Q_{\text{SH}}}. \quad (\text{R5})$$

Figure R3: An illustration of a programmable nonlinear microring resonator for a tunable SHG.

Assuming a loaded Q-factor of 30 million, we expect $P_{\text{th}} = 7.7 \mu\text{W}$ on a programmable SiN OPO.

Overall, we can see that a SiN programmable microring resonator has the potential to achieve state-of-the-art performance, even compared to more established, non-programmable platforms with much stronger native $\chi^{(2)}$ nonlinearities (e.g., lithium niobate with $\chi^{(2)} = 50 \text{ pm/V}$). This is due to the extremely low loss of SiN, which can fully compensate for the lower value of $\chi^{(2)}$ optical nonlinearity.

Potential Application 4: Programmable quantum light sources

Phase matching plays a crucial role in the application of biphoton generation using spontaneous parametric downconversion (SPDC). It determines various properties of the generated photon pair, such as wavelength, spatial profiles, and polarization [16]. Furthermore, nontrivial QPM grating structures can engineer more complex correlation structures of photons, enabling separable [39], highly multimodal [40], and even grid-like [41] correlations. Reference [33] provides a broader review of the potential of QPM engineering for SPDC light sources. The ability to programmably switch among these functions would enable highly flexible implementations of quantum technologies. For instance, similar to the concept proposed in Ref. [20], tunability of the wavelength of entangled photons would allow wavelength-division multiplexing in quantum networking, boosting the communication rate. There have been various demonstrations of tunable SPDC—e.g., via temperature tuning [42], angle tuning [43], or ferroelectric liquid crystals [44]—but engineering the full QPM grating structure remains elusive. To this end, a programmable nonlinear waveguide offers an ideal solution to these challenges. Below, we provide quantitative discussions on the potential performance of SPDC sources based on our approach.

A figure of merit often employed for the quantitative characterization of SPDC is the normalized brightness B_{norm} , which quantifies the number of photons generated per second per mW of pump power. For instance, $B_{\text{norm}} = 6.4 \times 10^4 \text{ pairs/s/mW}$ was measured for bulk beta barium borate (BBO) crystals, which are widely used in quantum optics experiments [45]. On a thin-film lithium niobate waveguide, $B_{\text{norm}} = 2.3 \times 10^{11} \text{ pairs/s/mW}$ was demonstrated [46]. Note that for many applications, a very high

Figure R4: Potential implementations of programmable SPDC on (a) a channel waveguide, (b) a waveguide array, and (c) a planar waveguide.

absolute count rate is neither necessary nor desirable, due to the undesired multi-photon events and the limitations in the speed of photon detectors and electronics. Thus, an important consideration is whether one can achieve the desired brightness with a reasonable pump power. Commercial SPDC sources operate around 1×10^4 pairs/s $\sim 1 \times 10^7$ pairs/s [47–49], implying the typical range of brightness required for applications.

For the estimation of realistic B_{norm} on programmable waveguides, we first consider a SiN programmable channel waveguide with $L = 0.7$ mm in length and an induced $\chi^{(2)} = 0.47$ pm/V—performance metrics we demonstrated in this work. The normalized brightness of a Type-0 degenerate SPDC process pumped by CW light in a waveguide is estimated as [50]

$$B_{\text{norm}} = \sqrt{\frac{2}{\pi^3} \frac{2}{3\epsilon_0 c^3} \frac{n_{g,a}^2}{n_a^4 n_b} \frac{d_{\text{eff}}^2 \omega_b^2}{\sqrt{|\beta_2|}} \left| \frac{\sigma_b^2}{\sigma_a^2 + 2\sigma_b^2} \right|^2} \frac{1}{\sigma_b^2} L^{3/2}, \quad (\text{R6})$$

where β_2 is the group-velocity dispersion at the FH wavelength, L is the length of the waveguide, and ω_b is the angular frequency of the pump (i.e., SH) light with wavelength 780 nm. For a programmable nonlinear waveguide, we have the effective nonlinearity $d_{\text{eff}} = \frac{1}{2\pi} \chi^{(2)}$. Here, the labels “a” and “b” denote the FH and SH modes, respectively. The formula assumes that both FH and SH have radially symmetric Gaussian beam profiles with beam radii of σ_a and σ_b , respectively. For the estimation of the SPDC rate, we assume the refractive indices of the FH and SH modes as $n_a = n_b = 2$, and the group index as $n_{g,a} = 2$. Assuming a moderate group-velocity dispersion of $|\beta_2| = 50$ fs²/mm, beam radii of $\sigma_a = \sigma_b = 1$ μm , and a total length of $L = 7$ mm, we find the normalized brightness of the device to be $B_{\text{norm}} = 8.4 \times 10^5$ pairs/s/mW—an order of magnitude greater than that of bulk BBO crystals. An absolute brightness of 1×10^7 pairs/s can be achieved with a reasonable pump power of 12 mW.

So far, we have focused our discussion on a programmable channel waveguide, but geometries with more spatial degrees of freedom offer intriguing potential for spatially multiplexed quantum light sources. In Fig. R4, we show illustrations of such extensions. For instance, various works have theoretically shown that SPDC occurring in an array of 1D waveguides can be used to produce a variety of spatial correlations of photons, such as cluster states, where pump shaping and engineered QPM gratings play central roles in determining the generated states [51–53]. Programmable SPDC on planar waveguide geometries could offer unique advantages as well. For example, it enables non-collinear generation of biphotons, as leveraged in Ref. [17], providing a simple way to separate entangled photon pairs. Additionally, free-form

propagation on a planar waveguide could offer comparable or even higher channel capacity and beam maneuverability [29]. A programmable waveguide can realize two-dimensional controllability of QPM gratings on such geometries, offering a unique opportunity for spatio-spectrally correlated programmable quantum light sources.

Below, in the boxes, we highlights new text to the manuscript that have been added to address the reviewer’s comment. While we added the full discussions on the potential applications 1-4 to the Supplementary Information, we do not show them here because they are nearly identical to parts of our reply above. Please refer to Appendix S2 for the corresponding text.

Box 10: Change in Sec. VII

Prospective applications

~~The ability to realize arbitrary $\chi^{(2)}(x, z)$ distributions makes our device platform very versatile, particularly in enabling devices that can seamlessly switch between performing multiple functions. In quantum technology, programmable nonlinear optics could enable a single physical device to perform multiple kinds of quantum gates, or quantum gates on qubits having different wavelengths [40, 54–56]. Similarly, all-optical signal processing for classical optical communications [57] could benefit from reconfigurable nonlinear processes, as could classical optical computation [58], and sensing with structured light [59–61]. Another possibility is to *augment* existing nanophotonic devices with programmable $\chi^{(2)}$ nonlinearity by adding a photoconductor layer and bias voltage. Finally, in situ inverse design may enable quantitative control over nonlinear optical processes for which we currently do not have accurate simulation models. Appendix S2 contains further discussion on potential applications.~~

The ability to realize arbitrary $\chi^{(2)}(x, z)$ distributions makes our device platform very versatile, particularly in enabling devices that must seamlessly switch between multiple functions. As shown in the demonstration of the programmable channel waveguide (see Appendix VIII B), our approach is, in principle, compatible with a wide range of device geometries beyond planer waveguides and could be seamlessly co-integrated with conventional photonic systems. Concretely, we have quantitatively analyzed the potential for our approach to be applied in four example application areas: (i) on-chip arbitrary pulse shapers, (ii) reconfigurable quantum frequency converters, (iii) widely wavelength-tunable integrated light sources, and (iv) quantum light sources with programmable entanglement structure (see Appendix S2). For these applications, we find that future SiN programmable nonlinear photonics could potentially achieve competitive or even state-of-art performance assuming the $\chi^{(2)}$ nonlinearity demonstrated in this work, all while allowing the flexibility enabled by programmability. In several of our application analyses, a crucial feature enabling programmable SiN to deliver competitive performance is the exceptionally low material loss of SiN, which enables snaking channel waveguides to be meters long and still have low loss [2, 18]. The long path lengths possible in SiN can fully compensate for the low $\chi^{(2)}$ nonlinear coefficient (relative to, for example, lithium niobate) and increase the programmable degrees of freedom. We expect that these four applications are not the only future possibilities: more speculatively, all-optical signal processing for classical optical communications [57] could benefit from reconfigurable nonlinear processes, as could classical optical computation [58], and sensing with structured light [59–61]. The scope of applications could further expand with the use of materials with higher inducible $\chi^{(2)}$ nonlinear-

ities [62]. Finally, in-situ inverse design may enable quantitative control over nonlinear-optical processes for which we currently do not have accurate simulation models.

Box 11: New text in the “Methods” section

Fabrication and characterizations of programmable nonlinear channel waveguides

In the main text, we primarily focused on a programmable planar waveguide. Although this design allowed us to engineer spatial features of light, it came with the drawback of lower light confinement, which reduced the conversion efficiency. One possible solution to this challenge is to fabricate a channel waveguide structure to prevent spatial diffraction, which enables light propagation over longer distances while maintaining tight transverse confinement. To demonstrate the compatibility of our approach with this channel waveguide geometry, we fabricated a prototypical programmable channel waveguide using SiN and characterized its performance.

The device was fabricated with the following steps. First, Silicon Valley Microelectronics provided a Si substrate with a 1 μm bottom oxide cladding and a 2 μm PECVD-grown SiN layer, identical to the substrate type used for the programmable planar waveguides. Next, we deposited a 500 nm-thick SiO₂ layer using PECVD, followed by sputtering a 200 nm-thick Cr layer. For photolithography, we spin-coated a deep ultraviolet (DUV) anti-reflective coating and photoresist. Exposure was performed using a DUV stepper (PAS 5500; ASML), after which we developed the photoresist and wet-etched the Cr to form a hard mask. We then etched through the SiN layer using CHF₃/O₂ gases in a plasma etcher (PlasmaPro 100 RIE; Oxford Instruments). To deposit additional cladding on the SiN sidewalls and thereby reduce losses into the photoconductive layer, we deposited 3 μm of SiO₂ with PECVD, and subsequently etched away 2.5 μm of oxide using the same plasma etching recipe described above. At this stage, we performed rapid thermal annealing at 800 °C for 5 mins. Finally, we deposited a 12 μm layer of SRN and a 20 nm layer of ITO as a transparent electrode. A cross-sectional scanning electron microscope (SEM) image of the fabricated channel waveguide is shown in Fig. 7. The SiN core was approximately 4 μm wide and 2 μm high. The waveguide had an overall length of 1.5 cm, with the programmable region limited to approximately 7 mm due to constraints of the imaging setup.

Fig. 7 An SEM image of the cross section of a programmable nonlinear channel waveguide. We highlight the boundaries between different materials with dashed lines.

We then conducted a nonlinear-optical characterization of the device. We coupled CW pump light with various wavelengths between 1540 nm and 1600 nm and measured the generated SHG while scanning the period of the monotonic QPM grating. We calculated the normalized SHG conversion efficiency by dividing the detected SHG power by the pump power squared measured after the waveguide, including appropriate corrections for propagation losses and collection inefficiencies (see Appendix S8 for details). The measurement results are shown in Fig. 8, which clearly demonstrate the capability to achieve phase matching at a desired pump wavelength by adjusting the QPM grating period.

Fig 8: Normalized SHG conversion efficiency of the programmable nonlinear channel waveguide for various pump wavelength (legend) and programmed QPM grating periods. Approximately 4 mW of pump power was detected after the waveguide. We used 1500 V of bias voltage with 10 Hz of frequency.

Note that our waveguide core measures approximately $2\ \mu\text{m} \times 4\ \mu\text{m}$ in cross section, supporting multiple spatial modes even at the pump wavelength. Moreover, the core’s skewed aspect ratio made it difficult to couple the pump light exclusively into the fundamental mode. Consequently, the pump power measured after the waveguide could include contributions from higher-order modes that do not contribute to SHG, which explains the wavelength-dependent variations in the normalized conversion efficiency. Thus, the efficiencies shown in Fig. 8 should be considered a lower bound of the device performance. The maximum normalized conversion efficiency observed was $\eta_{\text{norm}} = 2 \times 10^{-3} \text{ %/W}$, corresponding to a normalized slope conversion efficiency of $\eta_0 = 4 \times 10^{-3} \text{ %/W/cm}^2$, showing an approximately 40-fold improvement over the programmable nonlinear planar waveguide results presented in the main text. We note that reducing the core thickness below the micron scale to achieve single-mode operation causes the first-order poling period to fall below the smallest programmable feature size. If single-mode operation is prioritized, higher-order quasi-phase matching can be employed, albeit with reduced conversion efficiency. At the same time, the tighter field confinement increases nonlinear coupling and could compensate for the reduction of conversion efficiency.

Prospects for reducing the loss

There are various routes to reduce the absorption of SiN from the levels observed in Sec. S4 C1. In principle, SiN is capable of achieving extremely low optical loss. The material absorption limit of SiN has been estimated to be 0.13 dB/m, and losses as low as 0.4 dB/m have been demonstrated in high-confinement SiN waveguides [2,5]. These films were fabricated using low-pressure chemical vapor deposition (LPCVD). For PECVD SiN, it is known that residual hydrogen atoms in the film increase absorption, especially near a wavelength of 1520 nm [1,6]. Reduction of loss in PECVD SiN is possible either by using deuterated silane [3] or by high-temperature furnace annealing [4], both of which reduce the hydrogen concentration in the film. Here, we present preliminary experimental results using the latter approach.

For the annealing characterization, we fabricated channel waveguides using PECVD SiN. The fabrication procedure of the waveguide mostly followed the process we used for the programmable channel waveguide (see Sec. VIII B) with two differences; We stopped the process after the first plasma etching, and we replaced the rapid thermal annealing step with a 3 hours furnace anneal at 1200 °C with N₂ flow. The final die contained two types of waveguides: straight waveguides with a length of 2 cm and a spiral waveguide with a total length of 6 cm. In Fig. S15, we show a microscope image of the spiral waveguide.

Fig. S15: A microscope image of the SiN spiral waveguide. The image was created by stitching together nine individual images with smaller fields of view to capture the entire waveguide.

For the measurement of optical loss, we coupled a CW laser with a tunable wavelength between 1500 nm and 1630 nm into the waveguides and measured the power at the output. By dividing the ratio of output power by the difference in the lengths of the waveguides, we estimated the propagation loss per unit length. The measurement results are shown in Fig. S16. Though the data came out slightly noisy, potentially due to the multi-mode nature of the waveguide, we observed a clear reduction in optical loss—down to 0.4 dB/cm for wavelengths > 1550 nm and to 0.8 dB/cm around 1520 nm, where we had previously observed losses of 5 dB/cm. In the future, these loss values may be further improved by optimizing the etching recipe to reduce contributions from sidewall roughness.

Fig. S16: Optical loss of PECVD SiN waveguides after furnace annealing, estimated by comparing the power transmission through two waveguides of different lengths. We used a straight channel waveguide with a length of 2 cm and a spiral waveguide with a length of 6 cm.

Reviewer Comment 3.2 — *Finally, the manuscript comes with a 38-page supplement which, on the one hand, is great in terms of providing lots of details, but on the other hand is too much to expect most readers to go through. I request that the authors carefully consider what material is included in the supplement and ensure that their principal results and conclusions are sufficiently addressed in the main text, with the supplement used for backup/details but not any primary material. I also think that they need a table of contents and introductory section for the supplementary material so that it is easier for readers to navigate.*

Reply 3.2: We thank the reviewer for these ideas to improve the readability of our paper. Below, we describe how we have modified the manuscript to alleviate the issue of the long supplement being difficult to navigate, and we have reconsidered which details should appear in the main text versus in the supplement.

First, we have moved the following experimental details from the SI to the main text to minimize the need to refer to the SI:

- All figures in the main text include more experimental information, such as average pump power, pump pulse duration, and the applied bias voltage.
- We present the values of the optical loss of the waveguides with necessary background information.
- We have switched to absolute (non-normalized) units for all data in the main text so that readers can directly interpret the physical values of the signals.

Second, we have included the new results on the programmable channel waveguide in the “Methods” section instead of the SI in an attempt to ensure that this important information is easier to find than if it appeared in the supplement.

Finally, as shown in Box 13, we have added an introductory section and a table of contents to the SI to improve navigability.

Introduction

This document provides supplementary information related to the results presented in the main text. The sections in this document are organized into the following five categories:

- **Future prospects:** In Sec. S1, we summarize the current and potential future performance of the programmable nonlinear waveguides. In Sec. S2, we discuss potential applications of the platform based on quantitative metrics.
- **Characterization of the device:** These sections describe the physical parameters of the programmable nonlinear waveguides. The electrical and optical properties are summarized in Sec. S3 and Sec. S4, respectively. In Sec. S5, we provide details on how the programmable $\chi^{(2)}$ nonlinearity was estimated.
- **Model for the device operation:** In Sec. S6, we describe how to concisely model the nonlinear-optical behavior of the device for a given programming illumination.
- **Experimental details:** These sections provide additional information on the experimental results. The common parts of the setup are described in Sec. S7. The remaining sections cover CW-pumped SHG (Sec. S8), spectral engineering (Sec. S9), spatial engineering (Sec. S10), and spatio-spectral engineering (Sec. S11).
- **Theoretical background:** These sections offer supplementary theoretical context. In Sec. S12, we discuss the physics of electric-field-induced $\chi^{(2)}$ nonlinearity. In Sec. S13, we summarize methods for inverse designs in photonics.

Reviewer #4

Yanagimoto et al. report a fully programmable on-chip nonlinear photonic planar waveguide system capable of arbitrary reconfiguration through two-dimensional distribution of $\chi^{(2)}$ nonlinearity. This is achieved by patterned illumination $I(x,z)$ on a photoconductive layer, which induces a spatial electric field pattern within the waveguide core, thereby creating a 2D distribution of optical nonlinearity $\chi^{(2)}(x,z)$. As the authors fairly note, this work combines electric-field-induced $\chi^{(2)}$ (as in Refs. 12–18) with patterned illumination techniques (Refs. 44–45). While the mechanism of electric-field-induced $\chi^{(2)}$ has been well studied in previous years, a device with such strong programmable nonlinearity has not been demonstrated before. I believe this is a highly interesting and practically significant advancement in nonlinear photonic devices with high programmability.

Programmable linear optics has been well developed for classical and quantum photonic applications, but programmable nonlinearity remains relatively unexplored (previously, nonlinear crystals/sources were typically fixed, with reconfigurable pumping being the primary tuning method, e.g., Nat. Photonics 17, 573–581 (2023), which very differs from reconfiguring the nonlinear process itself). A relevant paper (Nat. Photon. 2025, <https://doi.org/10.1038/s41566-025-01660-x>)

was recently published, but it employs a very different approach to tuning nonlinearity. This work stands out due to its conceptual novelty and technological improvements, particularly in reducing linear loss in the electric-field-induced process and achieving substantial $\chi(2)$. The authors also provide a detailed discussion on further enhancing $\chi(2)$ to reach levels comparable to lithium niobate, addressing key factors such as bandgap wavelength (critical for SHG and SPDC), pattern resolution, and optical loss.

....This work pioneers a new direction in nonlinear optics and photonic devices, representing a milestone achievement in the field. With the above points addressed, I would be pleased to recommend its publication in Nature.

Reply 4.0: We thank the reviewer for the positive evaluation and accurate summary of the work.

Reviewer Comment 4.1 — Clarification of $\chi(2)(x,z)=3\chi(3)E_{\text{bias}}(x,z)$ is necessary. The derivation of this relationship is not entirely clear in the current manuscript. Given its importance in understanding SHG in third-order SiN materials, the authors should elaborate on whether the factor of three is dependent on the waveguide structures or materials.

Reply 4.1: We thank the reviewer for the comment. The factor of 3 in the expression $\chi^{(2)}(x,z) = 3\chi^{(3)}E_{\text{bias}}(x,z)$ arises from the algebraic expansion of the third-order nonlinear polarization. To clarify this point, we have added a derivation of this expression in the main text, as shown below.

Box 14: New text in Sec. II

The third-order nonlinear polarization induced by the sum of the bias field E_{bias} and the optical field E_{opt} (the field inside the waveguide, traveling in the z direction; not the programming illumination field) can be expanded as

$$P_{\text{NL}} = \chi^{(3)}(E_{\text{bias}} + E_{\text{opt}})^3 = \chi^{(3)}(E_{\text{opt}}^3 + 3E_{\text{bias}}E_{\text{opt}}^2 + 3E_{\text{bias}}^2E_{\text{opt}} + E_{\text{bias}}^3). \quad (\text{R7})$$

We can see that an effective quadratic nonlinearity arises as a term proportional to E_{opt}^2 , whose coefficient $\chi^{(2)}(x,z) = 3\chi^{(3)}E_{\text{bias}}(x,z)$ is proportional to the bias field. ~~This bias field in turn induced an effective $\chi^{(2)}$ optical nonlinearity via $\chi^{(2)}(x,z) = 3\chi^{(3)}E_{\text{bias}}(x,z)$.~~

Reviewer Comment 4.2 — The authors demonstrate flexible and comprehensive control over SHG dynamics, including spectral, spatial, and spatio-spectral (even real-time) tuning. A discussion on possible applications would strengthen the manuscript. For instance, static spectral and spatial control of QPM is highly relevant for engineering SPDC-based quantum light sources, enabling high-purity, indistinguishable single photons and multimode squeezed light (see conventional SPDC crystals from USTC's Pan group and waveguide sources from Paderborn's Silberhorn group). Additional insights into real-time dynamic control of QPM or nonlinearity could help envision future applications.

Reply 4.2: We thank the reviewer for the helpful suggestion. We agree with the reviewer that the potential for engineerable SPDC or squeezed-light generation is exciting and promising. We have now added quantitative analyses of four potential applications to the manuscript, including a quantum frequency converter (QFC) and quantum light sources based on spontaneous parametric downconversion (SPDC).

Below is the list of potential applications we examine:

- Potential Application 1: Highly programmable on-chip optical pulse shaper
- Potential Application 2: Programmable quantum frequency converter
- Potential Application 3: Widely tunable and highly efficient integrated light sources
- Potential Application 4: Programmable quantum light sources

In these sections, among other applications, we discuss how real-time control of the QPM grating could enable the quantum networking scheme recently proposed by Cisco Quantum Labs [20] via dense wavelength division multiplexing (WDM).

The boxes below show the changes in the text that have been incorporated to address the reviewer's comment.

Box 15: New text in Sec. VII

Prospective applications

~~The ability to realize arbitrary $\chi^{(2)}(x, z)$ distributions makes our device platform very versatile, particularly in enabling devices that can seamlessly switch between performing multiple functions. In quantum technology, programmable nonlinear optics could enable a single physical device to perform multiple kinds of quantum gates, or quantum gates on qubits having different wavelengths [40, 54–56]. Similarly, all-optical signal processing for classical optical communications [57] could benefit from reconfigurable nonlinear processes, as could classical optical computation [58], and sensing with structured light [59–61]. Another possibility is to *augment* existing nanophotonic devices with programmable $\chi^{(2)}$ nonlinearity by adding a photoconductor layer and bias voltage. Finally, in situ inverse design may enable quantitative control over nonlinear-optical processes for which we currently do not have accurate simulation models. Appendix S2 contains further discussion on potential applications.~~

The ability to realize arbitrary $\chi^{(2)}(x, z)$ distributions makes our device platform very versatile, particularly in enabling devices that must seamlessly switch between multiple functions. As shown in the demonstration of the programmable channel waveguide (see Appendix VIII B), our approach is, in principle, compatible with a wide range of device geometries beyond planer waveguides and could be seamlessly co-integrated with conventional photonic systems. Concretely, we have quantitatively analyzed the potential for our approach to be applied in four example application areas: (i) on-chip arbitrary pulse shapers, (ii) reconfigurable quantum frequency converters, (iii) widely wavelength-tunable integrated light sources, and (iv) quantum light sources with programmable entanglement structure (see Appendix S2). For these applications, we find that future SiN programmable nonlinear photonics could potentially achieve competitive or even state-of-art performance assuming the $\chi^{(2)}$ nonlinearity demonstrated in this work, all while allowing the flexibility enabled by programmability. In several of our application analyses, a crucial feature enabling programmable SiN to deliver competitive performance is the exceptionally low material loss of SiN, which enables snaking channel waveguides to be meters long and still have low loss [2, 18]. The long path lengths possible in SiN can fully compensate for the low $\chi^{(2)}$ nonlinear coefficient (relative to, for example, lithium niobate) and increase the programmable degrees of freedom. We expect that these four applications are not the only future possibilities: more speculatively, all-optical signal processing

for classical optical communications [57] could benefit from reconfigurable nonlinear processes, as could classical optical computation [58], and sensing with structured light [59–61]. The scope of applications could further expand with the use of materials with higher inducible $\chi^{(2)}$ nonlinearities [62]. Finally, in-situ inverse design may enable quantitative control over nonlinear-optical processes for which we currently do not have accurate simulation models.

Box 16: New text in Sec. S2 on programmable QFC

Programmable quantum frequency converter

A quantum frequency converter (QFC) is a device that converts the frequency of quantum-optical signals [31]. One of the major applications of QFCs is quantum networking, where the frequency of photons emitted from quantum nodes—often composed of atoms or vacancy centers—is converted to a frequency more suitable for long-range communication, and vice versa [32]. For this application, the ability to tune the target wavelengths is an essential feature of a QFC. It can compensate for the inherent inhomogeneities of the transition frequencies of certain quantum emitters and enables wavelength division multiplexing (WDM) to boost communication capacity. Recently, Cisco Quantum Labs published a roadmap for scalable quantum networking with dense WDM [20], where reconfigurable quantum interfaces—i.e., QFCs with tunable phase-matching conditions—play a central role. However, a suitable device has not yet been demonstrated. Our work could enable such an application, thanks to the fast and extremely wide programmability of the phase-matching conditions. More broadly, Ref. [33] reviews the unique role that nontrivial QPM structures can play in QFC applications, many of which can be implemented and dynamically switched on our platform. Below, we quantitatively show that a SiN programmable channel waveguide could realize high-efficiency conversion of quantum signals using sum/difference frequency (SFG/DFG) processes.

As illustrated in Fig. S2, we consider the same device design as discussed in Appendix S2 A but operated differently. The device consists of a programmable SiN channel waveguide with a slope SHG conversion efficiency of $\eta_0 = 4 \times 10^{-3} \text{ \%}/\text{W}/\text{cm}^2$ (as demonstrated in this work; see Sec. VIII B) and a total length of 1.5 m. We consider three different modes of light labeled as the “atomic mode”, “communication mode”, and “pump mode”, with frequencies ω_a , ω_c , and ω_p , respectively. The goal of a QFC is to convert photons in the atomic mode to the communication mode via a DFG process $\omega_a - \omega_p = \omega_c$. Note that this process also transduces photons in the communication mode to the atomic mode at the same time. Assigning annihilation operators to the atomic and communication modes as \hat{a} and \hat{c} , the DFG process induces the following transformation:

$$\hat{c}(L) = \cos(\sqrt{\eta_{0,\text{DFG}} P_p L}) \hat{c}(0) + \sin(\sqrt{\eta_{0,\text{DFG}} P_p L}) \hat{a}(0), \quad (\text{R8})$$

where $L = 1.5 \text{ m}$ is the length of the waveguide, P_p is the power of the pump field, and $\eta_{0,\text{DFG}}$ is the DFG slope conversion efficiency. Assuming weak wavelength dependence of the nonlinearity, $\eta_{0,\text{DFG}} = 4\eta_0$ holds, with the slope SHG conversion efficiency of the waveguide $\eta_0 = 4 \times 10^{-3} \text{ \%}/\text{W}/\text{cm}^2$. Full frequency conversion is achieved when $\hat{c}(L) = \hat{a}(0)$, corresponding to $P_p = 0.7 \text{ W}$. Such a power level can be straightforwardly achieved using quasi-CW pulsed operation. The normalized conversion efficiency of the device is $\eta_{0,\text{DFG}} \times (1.5 \text{ m})^2 = 360 \text{ \%}/\text{W}$, which exceeds the value $80 \text{ \%}/\text{W}$ of a commercially available QFC [34]. For a group-velocity mismatch of $0.4 \text{ ps}/\text{mm}$, the phase-matching bandwidth is expected to be 1.6 GHz . Such an extremely

narrow bandwidth, enabled by the long waveguide, can play the role of a filter, rejecting undesired background photons [35]. Overall, we expect the realization of such a programmable QFC to be feasible on SiN. The conversion efficiency η_0 we assumed was observed in our work. Furthermore, SiN waveguides longer than a meter have been demonstrated in various studies [18,19], and optical loss as low as 0.4 dB/m has been achieved [2]. An ability to reconfigure the QPM grating based on real-time experimental feedback would facilitate efficient quasi-phase matching on such a long waveguide.

Fig. S2 Illustration of a possible implementation of a QFC with a programmable nonlinear waveguide. The pump light drives a frequency conversion of a weak input quantum signal to the communication mode.

Box 17: New text in Sec. S2 on programmable SPDC

Programmable quantum light sources

Phase matching plays a crucial role in the application of biphoton generation using spontaneous parametric downconversion (SPDC). It determines various properties of the generated photon pair, such as wavelength, spatial profiles, and polarization [16]. Furthermore, nontrivial QPM grating structures can engineer more complex correlation structures of photons, enabling separable [39], highly multimodal [40], and even grid-like [41] correlations. Reference [33] provides a broader review of the potential of QPM engineering for SPDC light sources. The ability to programmably switch among these functions would enable highly flexible implementations of quantum technologies. For instance, similar to the concept proposed in Ref. [20], tunability of the wavelength of entangled photons would allow wavelength-division multiplexing in quantum networking, boosting the communication rate. There have been various demonstrations of tunable SPDC—e.g., via temperature tuning [42], angle tuning [43], or ferroelectric liquid crystals [44]—but engineering the full

QPM grating structure remains elusive. To this end, a programmable nonlinear waveguide offers an ideal solution to these challenges. Below, we provide quantitative discussions on the potential performance of SPDC sources based on our approach.

Fig. S4 Illustration of possible future implementations of programmable SPDC on (a) a channel waveguide, (b) a waveguide array, and (c) a planar waveguide.

A figure of merit often employed for quantitative characterizations of SPDC is the normalized brightness B_{norm} , which quantifies the number of photons generated per second per mW of pump power. For instance, $B_{\text{norm}} = 6.4 \times 10^4$ pairs/s/mW was measured for bulk beta barium borate (BBO) crystals, which are widely used in quantum optics experiments [45]. On a thin-film lithium niobate waveguide, $B_{\text{norm}} = 2.3 \times 10^{11}$ pairs/s/mW was demonstrated [46]. Note that for many applications, a very high absolute count rate is neither necessary nor desirable, due to the undesired multi-photon events and the limitations in the speed of photon detectors and electronics. Thus, an important consideration is whether one can achieve the desired brightness with a reasonable pump power. Commercial SPDC sources operate around 1×10^4 pairs/s $\sim 1 \times 10^7$ pairs/s [47–49], implying the typical range of brightness required for applications.

For the estimation of realistic B_{norm} on programmable waveguides, we first consider a SiN programmable channel waveguide with $L = 0.7$ mm in length and an induced $\chi^{(2)} = 0.47$ pm/V—performance metrics we demonstrated in this work. The normalized brightness of a Type-0 degenerate SPDC process pumped by CW light in a waveguide is estimated as [50]

$$B_{\text{norm}} = \sqrt{\frac{2}{\pi^3}} \frac{2}{3\epsilon_0 c^3} \frac{n_{g,a}^2}{n_a^4 n_b} \frac{d_{\text{eff}}^2 \omega_b^2}{\sqrt{|\beta_2|}} \left| \frac{\sigma_b^2}{\sigma_a^2 + 2\sigma_b^2} \right|^2 \frac{1}{\sigma_b^2} L^{3/2}, \quad (\text{R9})$$

where β_2 is the group-velocity dispersion at the FH wavelength, L is the length of the waveguide, and ω_b is the angular frequency of the pump (i.e., SH) light with wavelength 780 nm. For a programmable nonlinear waveguide, we have the effective nonlinearity $d_{\text{eff}} = \frac{1}{2\pi} \chi^{(2)}$. Here, the labels “a” and “b” denote the FH and SH modes, respectively. The formula assumes that both FH and SH have radially symmetric Gaussian beam profiles with beam radii of σ_a and σ_b , respectively. For the estimation of the SPDC rate, we assume the refractive indices of the FH and SH modes as $n_a = n_b = 2$, and the group index as $n_{g,a} = 2$. Assuming a moderate group-velocity dispersion of $|\beta_2| = 50 \text{ fs}^2/\text{mm}$, beam radii of $\sigma_a = \sigma_b = 1 \mu\text{m}$, and a total length of $L = 7 \text{ mm}$, we find the

normalized brightness of the device to be $B_{\text{norm}} = 8.4 \times 10^5$ pairs/s/mW—an order of magnitude greater than that of bulk BBO crystals. An absolute brightness of 1×10^7 pairs/s can be achieved with a reasonable pump power of 12 mW.

So far, we have focused our discussion on a programmable channel waveguide, but geometries with more spatial degrees of freedom offer intriguing potential for spatially multiplexed quantum light sources. In Fig. S4, we show illustrations of such extensions. For instance, various works have theoretically shown that SPDC occurring in an array of 1D waveguides can be used to produce a variety of spatial correlations of photons, such as cluster states, where pump shaping and engineered QPM gratings play central roles in determining the generated states [51–53]. Programmable SPDC on planar waveguide geometries could offer unique advantages as well. For example, it enables non-collinear generation of biphotons, as leveraged in Ref. [17], providing a simple way to separate entangled photon pairs. Additionally, free-form propagation on a planar waveguide could offer comparable or even higher channel capacity and beam maneuverability [29]. A programmable waveguide can realize two-dimensional controllability of QPM gratings on such geometries, offering a unique opportunity for programmable quantum light sources with spatio-spectral controls of entanglement structures.

Reviewer Comment 4.3 — *The authors mention that the applied electric field E_{bias} must remain below the material’s breakdown threshold. However, potential breakdown due to the external pump light and illumination light on the photoconductive layer and in the core layer should also be addressed. Another critical aspect is the lifetime of the patterned $\chi^{(2)}$ distribution. How stable is it after applying the electric field and illumination? Its resilience under environmental factors (e.g., temperature fluctuations) should be clarified.*

Reply 4.3: We thank the reviewer for the comment. Regarding breakdown due to the pump light, Ref. [30] evaluated the breakdown of SiN waveguides using femtosecond pulses and found a damage threshold of 7.6 GW/mm^2 . For a planar waveguide with a beam cross-sectional area of approximately $2 \mu\text{m} \times 150 \mu\text{m}$, this corresponds to 2 MW of peak power, which is far above the maximum output of the pump laser we used in the experiment (ELMO-HP with approximately 30 kW peak power off chip). For the programmable channel waveguide, which we newly fabricated during the revision of the manuscript with a cross-sectional area of $2 \mu\text{m} \times 4 \mu\text{m}$, the damage threshold is 60 kW. The optical power of the illumination light used in this work was less than 100 mW/cm^2 , which is sufficiently low to ensure that neither the core nor the photoconductor layer is damaged. We therefore expect that breakdown due to the programming illumination is unlikely as long as the intensity remains at this level.

To test the long-term stability of the $\chi^{(2)}$ distribution, we conducted continuous measurements of pulsed SHG output as shown in Fig. S28, where a sequence of QPM grating patterns is repeatedly projected. By measuring the power fluctuations of the SHG with the same QPM grating pattern, we observed that the power fluctuated around 15% over 10 hours of operation. However, such fluctuations may be caused by alignment drift of the pump light from the fundamental mode of the waveguide, due to mechanical vibrations and a few degrees of temperature fluctuation in the room. Contributions from this effect are currently difficult to calibrate due to the multimode nature of the waveguide. We therefore interpret the SHG power fluctuation as an upper bound for the fluctuation in the $\chi^{(2)}$ nonlinearity, which we estimate to be approximately 7% over 10 hours under a few degrees of temperature fluctuation. Despite the overall power fluctuations, the complicated spectral features (i.e., “CORNELL” pattern)

in the SHG time trace remained clear throughout the measurements, underscoring the stability of the phase-matching functions.

We have incorporated these discussions into the manuscript, as shown in the boxes below.

Box 18: New text in Sec. S6

The illumination intensity on the chip surface is approximately 50 mW/cm^2 , which is low enough to avoid causing breakdown damage to the films.

Box 19: New text in Sec. S4

Optical damage threshold

Here, we comment on the amount of power that the programmable nonlinear waveguides can handle, which provides a rough estimate of the range of applications for which the device may be useful. Using 250 fs ultrashort pulses, Ref. [30] reported a damage threshold of 0.19 J/cm^2 , corresponding to 7.6 GW/mm^2 . For a SiN planar waveguide with a cross-sectional mode area of $2 \mu\text{m} \times 150 \mu\text{m}$, the damage threshold corresponds to 2.3 MW. At half this damage threshold and using the normalized conversion efficiency of $\eta_{\text{norm}} = 5 \times 10^{-5} \text{ \%}/\text{W}$, a value demonstrated in this work, the total conversion efficiency would be $\eta_{\text{tot}} = 6 \times 10^1 \text{ \%}$. For the programmable channel waveguide shown in Sec. VIII B, the approximate cross-sectional area is $2 \mu\text{m} \times 4 \mu\text{m}$, leading to a damage threshold of 61 kW. With the observed normalized conversion efficiency of $\eta_{\text{norm}} = 2 \times 10^{-3} \text{ \%}/\text{W}$ and operating at half of the damage threshold, the total conversion efficiency becomes $\eta_{\text{tot}} = 6 \times 10^1 \text{ \%}$.

Box 20: New text in Sec. S8

In Fig. 3(d), we updated the programming illumination pattern $I(x, z)$ in real time, effectively projecting a “movie” onto the surface of a programmable nonlinear waveguide to achieve dynamic control of the broadband SHG spectrum. Although we show only approximately $\sim 300 \text{ s}$ of the trace in the main text, the operation was highly stable and could continue for much longer periods. In Fig. S28, we present a time trace of the SHG spectrum over 10 hours of operation, during which numerous “Cornell” patterns were generated. This process involved reconfiguring the $\chi^{(2)}$ nonlinearity 15,400 times, and we did not observe any practical upper bound on the cycle count. These results provide compelling visual evidence of the stability and repeatability of the programmable nonlinear waveguide. By comparing the SHG power from the same QPM pattern over the course of the measurement, we observed approximately 15% drift in power. Although such fluctuations may be caused by misalignment of the pump beam from the fundamental mode of the waveguide—e.g., due to typical temperature fluctuations between $21 \text{ }^\circ\text{C}$ and $23 \text{ }^\circ\text{C}$ in the room or mechanical vibrations—it is challenging to accurately estimate their contribution due to the multimode nature of the waveguide. This measurement provides an upper bound on the instability of the $\chi^{(2)}$ nonlinearity, estimated to be approximately 7% over 10 hours under $2 \text{ }^\circ\text{C}$ degrees of temperature variation.

Reviewer Comment 4.4 — *The current planar waveguide features a $2 \mu\text{m}$ SiN core with $1 \mu\text{m}$ SiO₂ cladding, supporting multimodes at both pump and SHG wavelengths. Could the authors discuss the*

feasibility of reducing the core thickness to achieve single-mode operation (vertically)? Etched ridge waveguides, which confine light to a single mode in both vertical and horizontal directions, are more relevant for photonic integration. Would the proposed nonlinear photonic waveguide be adaptable to such single-mode, 1D reconfigurable structures? This is of significant practical importance.

Reply 4.4: We thank the reviewer for the comment and question on the potential realizations of single-mode operation and etched channel waveguides. To demonstrate that our approach indeed works well on nanophotonic waveguides with tighter field confinement, we fabricated a programmable nonlinear *channel* waveguide during the revision of the manuscript. As shown in Box 21 below, we observed a clear signature of programmable quasi-phase matching, as well as a 40-fold improvement in the normalized SHG conversion efficiency. Therefore the answer to whether our approach can be adapted to make 1D reconfigurable structures is happily: yes.

At the same time, the cross-sectional area of the channel waveguide we newly fabricated was $2\ \mu\text{m} \times 4\ \mu\text{m}$, which is still not small enough to fully ensure single-mode operation. While it is, in principle, possible to reduce the mode cross-section further, doing so generally makes the phase-matching condition more stringent. Due to the current resolution limit of the programmable features, we found it challenging to reduce the size of the waveguide below the sub-micron scale while maintaining the ability to perform first-order quasi-phase matching. If an application of interest prioritizes single-mode operation, one could employ higher-order quasi-phase matching, albeit with reduced conversion efficiency. At the same time, the tighter field confinement increases nonlinear coupling, and this could partially compensate for the reduction of conversion efficiency.

A discussion of these aspects of engineering 1D reconfigurable waveguides is included in the new text shown in Box 21. Other methods to realize finer programmable features are presented in Appendix S3 C.

Box 21: New text in the “Methods” section

Fabrication and characterizations of programmable nonlinear channel waveguides

In the main text, we primarily focused on a programmable planar waveguide. Although this design allowed us to engineer spatial features of light, it came with the drawback of lower light confinement, which reduced the conversion efficiency. One possible solution to this challenge is to fabricate a channel waveguide structure to prevent spatial diffraction, which enables light propagation over longer distances while maintaining tight transverse confinement. To demonstrate the compatibility of our approach with this channel waveguide geometry, we fabricated a prototypical programmable channel waveguide using SiN and characterized its performance.

The device was fabricated with the following steps. First, Silicon Valley Microelectronics provided a Si substrate with a $1\ \mu\text{m}$ bottom oxide cladding and a $2\ \mu\text{m}$ PECVD-grown SiN layer, identical to the substrate type used for the programmable planar waveguides. Next, we deposited a $500\ \text{nm}$ -thick SiO_2 layer using PECVD, followed by sputtering a $200\ \text{nm}$ -thick Cr layer. For photolithography, we spin-coated a deep ultraviolet (DUV) anti-reflective coating and photoresist. Exposure was performed using a DUV stepper (PAS 5500; ASML), after which we developed the photoresist and wet-etched the Cr to form a hard mask. We then etched through the SiN layer using CHF_3/O_2 gases in a plasma etcher (PlasmaPro 100 RIE; Oxford Instruments). To deposit additional cladding on the SiN sidewalls and thereby reduce losses into the photoconductive layer, we deposited $3\ \mu\text{m}$ of SiO_2 with PECVD, and subsequently etched away $2.5\ \mu\text{m}$ of oxide using the

same plasma etching recipe described above. At this stage, we performed rapid thermal annealing at 800 °C for 5 mins. Finally, we deposited a 12 μm layer of SRN and a 20 nm layer of ITO as a transparent electrode. A cross-sectional scanning electron microscope (SEM) image of the fabricated channel waveguide is shown in Fig. 7. The SiN core was approximately 4 μm wide and 2 μm high. The waveguide had an overall length of 1.5 cm, with the programmable region limited to approximately 7 mm due to constraints of the imaging setup.

Fig. 7 An SEM image of the cross section of a programmable nonlinear channel waveguide. We highlight the boundaries between different materials with dashed lines.

We then conducted a nonlinear-optical characterization of the device. We coupled CW pump light with various wavelengths between 1540 nm and 1600 nm and measured the generated SHG while scanning the period of the monotonic QPM grating. We calculated the normalized SHG conversion efficiency by dividing the detected SHG power by the pump power squared measured after the waveguide, including appropriate corrections for propagation losses and collection inefficiencies (see Appendix S8 for details). The measurement results are shown in Fig. 8, which clearly demonstrate the capability to achieve phase matching at a desired pump wavelength by adjusting the QPM grating period.

Fig 8: Normalized SHG conversion efficiency of the programmable nonlinear channel waveguide for various pump wavelength (legend) and programmed QPM grating periods. Approximately 4 mW of pump power was detected after the waveguide. We used 1500 V of bias voltage with 10 Hz of frequency.

Note that our waveguide core measures approximately $2\ \mu\text{m} \times 4\ \mu\text{m}$ in cross section, supporting multiple spatial modes even at the pump wavelength. Moreover, the core's skewed aspect ratio made it difficult to couple the pump light exclusively into the fundamental mode. Consequently, the pump power measured after the waveguide could include contributions from higher-order modes that do not contribute to SHG, which explains the wavelength-dependent variations in the normalized conversion efficiency. Thus, the efficiencies shown in Fig. 8 should be considered a lower bound of the device performance. The maximum normalized conversion efficiency observed was $\eta_{\text{norm}} = 2 \times 10^{-3} \text{ \%}/\text{W}$, corresponding to a normalized slope conversion efficiency of $\eta_0 = 4 \times 10^{-3} \text{ \%}/\text{W}/\text{cm}^2$, showing an approximately 40-fold improvement over the programmable nonlinear planar waveguide results presented in the main text. We note that reducing the core thickness below the micron scale to achieve single-mode operation causes the first-order poling period to fall below the smallest programmable feature size. If single-mode operation is prioritized, higher-order quasi-phase matching can be employed, albeit with reduced conversion efficiency. At the same time, the tighter field confinement increases nonlinear coupling and could compensate for the reduction of conversion efficiency.

Reviewer Comment 4.5 — *Related comment: The authors note the sensitivity of 1D PPLN structures to fabrication variations, necessitating adaptive poling. However, the primary reason for PPLN sensitivity lies in thin-film LN layers (hundreds of nanometers thick); thicker layers are far less susceptible (see conventional bulk PPLN waveguides). The authors should clarify their statement.*

Reply 4.5: We thank the reviewer for the comment on the sensitivity to poling noise. It is indeed true that devices with larger mode volumes are much less sensitive to poling noise, and thus, adaptive poling was not necessary for achieving efficient phase matching in the prototypical devices presented in this work. However, we expect that real-time adaptive poling will be valuable for future devices with more stringent requirements for periodic poling. For instance, the meters-long waveguides considered for various potential applications in Appendix S2 would greatly benefit from such real-time programmability.

Below, we show the portion of the text that has been modified to address the comment.

Box 22: New text in Sec. S2

100 %-yield QPM gratings

For example, nanoscale thickness variations in thin-film lithium niobate (TFLN) waveguides typically place an effective limit on the useful length of a periodically poled waveguide and the maximum achievable conversion efficiency. In Ref. [63], the authors reported how they could circumvent this limitation by precisely measuring the thickness distribution and adapting the poling to compensate for the thickness inhomogeneity. Programmable nonlinear photonic devices offer a fundamentally different solution to such challenges. By dynamically optimizing the QPM grating structure for an

experimentally measured figure of merit (FOM), programmable devices can be adapted in real time to maximize performance. We demonstrated both robustness to fluctuations in pump wavelength (Sec. III), where the FOM was conversion efficiency, and in situ inverse design (Sec. IV), where the FOM was the similarity between the measured and target SHG spectra. *We note that the prototype devices demonstrated in in this work had a much thicker film and lower field confinement than the aforementioned TFLN waveguides, making them less sensitive to poling errors. Thus, such real-time adaptive poling was not technically necessary to achieve efficient phase-matching conditions over lengths of less than a centimeter. Rather, these demonstrations should be viewed as proofs of concept for futuristic devices with more stringent requirements for the accuracy of periodic poling, e.g., meters-long programmable waveguides as discussed in Appendix S2 A and Appendix S2 B.* Furthermore, there are many situations in nonlinear optics—supercontinuum generation being a prominent example [64]—where device behavior is extremely sensitive not only to the device parameters but also to the field profile of the pump light, making it challenging to achieve exact agreement between simulation and experiment. The inverse-design experiments reported in Sec. IV were performed without prior characterization of the pump; in situ inverse design using programmable devices may ultimately enable the realization of quantitatively correct behavior even for complex nonlinear-optical processes that we don't have accurate simulation models for.

Reviewer Comment 4.6 — *Could the authors also comment on how the current (or improved) nonlinear waveguides might be integrated with conventional photonic waveguides for system-level applications?*

Reply 4.6: We thank the reviewer for the comment. Because the photoconductive layer is separated from the layers beneath and—in our experiments—appear to not cause adverse effects, we expect the system-level integration of our approach with conventional photonic systems to be relatively straightforward. This point is reinforced by the newly added results to the manuscript (See Sec. VIII), where we demonstrated that programmable nonlinearity could be augmented to a channel waveguide without adversely affecting the waveguide.

Beyond the systems-integration concern about the introduction of a photoconductor layer, we can also give our thoughts on the systems-integration possibilities for a device based on a planar waveguide rather than the channel waveguide structures used in many systems: There have been various demonstrations in which two-dimensional planar waveguides—often with inversely designed etched structures—are connected to arrays of conventional channel waveguides, which serve as input/output ports [65–68]. These systems can simultaneously benefit from the two-dimensional engineerability of the planar waveguide while maintaining integrability with the rest of the system. We expect that programmable nonlinear planar waveguides could be integrated using the same approach, where spatial engineering can be used to route beams to desired output ports in a reconfigurable manner. The SiN channel waveguides, either programmable or non-programmable, could then be efficiently interconnected with other materials, such as III–V semiconductors and lithium niobate [69, 70]. Such heterogeneous integration could further expand the functional scope of the system.

For instance, one could consider integrating a conventional microcomb to directly pump a pulse shaper implemented using a programmable nonlinear waveguide (see Appendix S2 A for details on the pulse shaper). This would enable a compact system that converts CW input pump light into a train of mode-locked optical pulses with programmable spectral–temporal profiles. Such a system could be

further extended to pump subsequent photonic waveguides—e.g., for spectro-temporally engineered quantum light sources [40]—thereby realizing a programmable broadband quantum light source.

This is not to say that it will be trivial to engineer end-to-end systems incorporating the our programmable nonlinearity approach: certainly some further research will be needed to establish the methods for integrating our approach with various conventional photonic systems. However, we see no fundamental obstacles to performing systems integration and believe it will be similarly difficult to the usual systems-integration challenges for non-programmable systems.

We have incorporated these statements regarding system-level integration into the corresponding parts of the text, as shown below.

Box 23: New text in Sec. S2 A

With programmable illumination implemented using micro-LED arrays [29] and pump light source provided by micro-combs, the entire pulse shaper setup could be fully integrated on-chip.

Box 24: New text in Sec. VII

As shown in the demonstration of the programmable channel waveguide (see Appendix VIII B), our approach is, in principle, compatible with a wide range of device geometries beyond planer waveguides and could be seamlessly co-integrated with conventional photonic systems.

Reviewer Comment 4.7 — *Minor corrections: The color scheme for SiO₂ cladding is inconsistent between Fig. 1a (gray) and Fig. 1b/Fig.6 (pink), which may cause confusion. The x-axis label is missing in Fig. 4a.*

Reply 4.7: We thank the reviewer for pointing out the issues in the figures. We have corrected the inconsistency in the color scheme of SiO₂ and added x-axis label to Fig. 4a.

Additional edits

During the revision of the manuscript, we realized that there were errors in the code used to produce the figures presenting the time traces of the SHG spectrum (Figures 3(c) and S29 in the revised manuscript). Instead of showing the SHG spectrum acquired in real time for dynamically changing QPM gratings, the code was plotting the spectrum measured beforehand for each QPM grating as a reference. We have fixed the figures to represent the correct experimental results. This change does not affect the statements in the manuscript.

References

- [1] Wang, L., Xie, W., Van Thourhout, D., Zhang, Y., Yu, H. & Wang, S. Nonlinear silicon nitride waveguides based on pecvd deposition platform. *Optics Express* **26**, 9645 (2018).
- [2] Ji, X., Roberts, S., Corato-Zanarella, M. & Lipson, M. Methods to achieve ultra-high quality factor silicon nitride resonators. *APL Photonics* **6** (2021).

- [3] Bose, D. et al. Anneal-free ultra-low loss silicon nitride integrated photonics. *Light: Science & Applications* **13**, 156 (2024).
- [4] Ji, X., Okawachi, Y., Gil-Molina, A., Corato-Zanarella, M., Roberts, S., Gaeta, A. L. & Lipson, M. Ultra-low-loss silicon nitride photonics based on deposited films compatible with foundries. *Laser & Photonics Reviews* **17**, 2200544 (2023).
- [5] Ji, X. et al. Ultra-low-loss on-chip resonators with sub-milliwatt parametric oscillation threshold. *Optica* **4**, 619–624 (2017).
- [6] Ay, F. & Aydinli, A. Comparative investigation of hydrogen bonding in silicon based pecvd grown dielectrics for optical waveguides. *Optical Materials* **26**, 33–46 (2004).
- [7] Shiloh, R. & Arie, A. Spectral and temporal holograms with nonlinear optics. *Optics Letters* **37**, 3591–3593 (2012).
- [8] Leshem, A., Shiloh, R. & Arie, A. Experimental realization of spectral shaping using nonlinear optical holograms. *Optics Letters* **39**, 5370–5373 (2014).
- [9] Billat, A., Grassani, D., Pfeiffer, M. H. P., Kharitonov, S., Kippenberg, T. J. & Brès, C.-S. Large second harmonic generation enhancement in si₃n₄ waveguides by all-optically induced quasi-phase-matching. *Nature Communications* **8** (2017).
- [10] Lu, X., Moille, G., Rao, A., Westly, D. A. & Srinivasan, K. Efficient photoinduced second-harmonic generation in silicon nitride photonics. *Nature Photonics* **15**, 131–136 (2020).
- [11] Hickstein, D. D. et al. Self-organized nonlinear gratings for ultrafast nanophotonics. *Nature Photonics* **13**, 494–499 (2019).
- [12] Nitiss, E., Hu, J., Stroganov, A. & Brès, C.-S. Optically reconfigurable quasi-phase-matching in silicon nitride microresonators. *Nature Photonics* **16**, 134–141 (2022).
- [13] Li, B. et al. Down-converted photon pairs in a high-q silicon nitride microresonator. *Nature* (2025).
- [14] Wang, G. et al. Integrated tunable green light source on silicon nitride. *arXiv preprint arXiv:2504.13662* (2025).
- [15] Yuan, Z. et al. Efficient and wavelength-tunable second-harmonic generation toward the green gap. *Science Advances* **11**, eadw2781 (2025).
- [16] Couteau, C. Spontaneous parametric down-conversion. *Contemporary Physics* **59**, 291–304 (2018).
- [17] Cao, B. et al. Non-collinear generation of ultra-broadband parametric fluorescence photon pairs using chirped quasi-phase matching slab waveguides. *Optics Express* **31**, 23551–23562 (2023).
- [18] Liu, J. et al. High-yield, wafer-scale fabrication of ultralow-loss, dispersion-engineered silicon nitride photonic circuits. *Nature communications* **12**, 2236 (2021).

- [19] Bauters, J. F. et al. Planar waveguides with less than 0.1 db/m propagation loss fabricated with wafer bonding. *Optics express* **19**, 24090–24101 (2011).
- [20] Zhao, J. et al. Scalable mhz-rate entanglement distribution in low-latency quantum networks interconnecting heterogeneous quantum processors. *arXiv preprint arXiv:2504.05567* (2025).
- [21] Weiner, A. M. Ultrafast optical pulse shaping: A tutorial review. *Optics Communications* **284**, 3669–3692 (2011).
- [22] Liu, Y., Park, S.-G. & Weiner, A. Terahertz waveform synthesis via optical pulse shaping. *IEEE Journal of Selected Topics in Quantum Electronics* **2**, 709–719 (1996).
- [23] Jiang, Z., Leaird, D. & Weiner, A. Line-by-line pulse shaping control for optical arbitrary waveform generation. *Optics Express* **13**, 10431–10439 (2005).
- [24] http://www.optophase.com/images/Biophotonic%20solutions/MIIPSTBox640_sheet.pdf. Accessed: 2025-07-08.
- [25] Divitt, S., Zhu, W., Zhang, C., Lezec, H. J. & Agrawal, A. Ultrafast optical pulse shaping using dielectric metasurfaces. *Science* **364**, 890–894 (2019).
- [26] <https://amplitude-laser.com/products/femtosecond-lasers/instrumentation-lasers-femtosecondes/dazzler/>. Accessed: 2025-07-08.
- [27] Metcalf, A. J. et al. Integrated line-by-line optical pulse shaper for high-fidelity and rapidly reconfigurable rf-filtering. *Optics Express* **24**, 23925–23940 (2016).
- [28] Imeshev, G., Galvanauskas, A., Harter, D., Arbore, M. A., Proctor, M. & Fejer, M. M. Engineerable femtosecond pulse shaping by second-harmonic generation with fourier synthetic quasi-phase-matching gratings. *Optics Letters* **23**, 864 (1998).
- [29] Onodera, T. et al. Scaling on-chip photonic neural processors using arbitrarily programmable wave propagation, <https://arxiv.org/abs/2402.17750> (2024).
- [30] Tan, S. et al. Silicon nitride waveguide as a power delivery component for on-chip dielectric laser accelerators. *Optics letters* **44**, 335–338 (2019).
- [31] Kumar, P. Quantum frequency conversion. *Optics letters* **15**, 1476–1478 (1990).
- [32] Wei, S.-H. et al. Towards real-world quantum networks: a review. *Laser & Photonics Reviews* **16**, 2100219 (2022).
- [33] Weiss, T. F. & Peruzzo, A. Nonlinear domain engineering for quantum technologies. *Applied Physics Reviews* **12** (2025).
- [34] <https://quantumcomputinginc.com/products/commercial-products/frequency-converter>. Accessed: 2025-07-10.
- [35] Sua, Y. M., Fan, H., Shahverdi, A., Chen, J.-Y. & Huang, Y.-P. Direct generation and detection of quantum correlated photons with 3.2 um wavelength spacing. *Scientific reports* **7**, 17494 (2017).

- [36] Lu, J., Li, M., Zou, C.-L., Al Sayem, A. & Tang, H. X. Toward 1% single-photon anharmonicity with periodically poled lithium niobate microring resonators. *Optica* **7**, 1654–1659 (2020).
- [37] Lu, J., Al Sayem, A., Gong, Z., Surya, J. B., Zou, C.-L. & Tang, H. X. Ultralow-threshold thin-film lithium niobate optical parametric oscillator. *Optica* **8**, 539–544 (2021).
- [38] Yanagimoto, R., Ng, E., Jankowski, M., Mabuchi, H. & Hamerly, R. Temporal trapping: a route to strong coupling and deterministic optical quantum computation. *Optica* **9**, 1289–1296 (2022).
- [39] Xin, C. J. et al. Spectrally separable photon-pair generation in dispersion engineered thin-film lithium niobate. *Optics Letters* **47**, 2830 (2022).
- [40] Ansari, V., Donohue, J. M., Brecht, B. & Silberhorn, C. Tailoring nonlinear processes for quantum optics with pulsed temporal-mode encodings. *Optica* **5**, 534–550 (2018).
- [41] Zhu, J.-L., Zhu, W.-X., Shi, X.-T., Zhang, C.-T., Hao, X., Yang, Z.-X. & Jin, R.-B. Design of mid-infrared entangled photon sources using lithium niobate. *Journal of the Optical Society of America B* **40**, A9–A16 (2023).
- [42] Fedrizzi, A., Herbst, T., Poppe, A., Jennewein, T. & Zeilinger, A. A wavelength-tunable fiber-coupled source of narrowband entangled photons. *Optics Express* **15**, 15377–15386 (2007).
- [43] Shen, L., Lee, J., Hartanto, A. W., Tan, P. & Kurtsiefer, C. Wide-range wavelength-tunable photon-pair source for characterizing single-photon detectors. *Optics Express* **29**, 3415–3424 (2021).
- [44] Sultanov, V., Kavčič, A., Kokkinakis, E., Sebastián, N., Chekhova, M. V. & Humar, M. Tunable entangled photon-pair generation in a liquid crystal. *Nature* **631**, 294–299 (2024).
- [45] Villar, A., Lohrmann, A. & Ling, A. Experimental entangled photon pair generation using crystals with parallel optical axes. *Optics express* **26**, 12396–12402 (2018).
- [46] Harper, N. A., Hwang, E. Y., Sekine, R., Ledezma, L., Perez, C., Marandi, A. & Cushing, S. K. Highly efficient visible and near-ir photon pair generation with thin-film lithium niobate. *Optica Quantum* **2**, 103–109 (2024).
- [47] https://www.thorlabs.com/newgrouppage9.cfm?objectgroup_id=13675. Accessed: 2025-07-10.
- [48] https://www.ozoptics.com/ALLNEW_PDF/DTS0199.pdf. Accessed: 2025-07-10.
- [49] <https://s-fifteen.com/products/cpps-810-correlated-photon-pair-source>. Accessed: 2025-07-10.
- [50] Schneeloch, J. et al. Introduction to the absolute brightness and number statistics in spontaneous parametric down-conversion. *Journal of Optics* **21**, 043501 (2019).
- [51] Solntsev, A. S., Sukhorukov, A. A., Neshev, D. N. & Kivshar, Y. S. Spontaneous parametric down-conversion and quantum walks in arrays of quadratic nonlinear waveguides. *Physical Review Letters* **108**, 023601 (2012).

- [52] Titchener, J. G., Solntsev, A. S. & Sukhorukov, A. A. Reconfigurable cluster-state generation in specially poled nonlinear waveguide arrays. *Physical Review A* **101**, 023809 (2020).
- [53] Barral, D., Walschaers, M., Bencheikh, K., Parigi, V., Levenson, J. A., Treps, N. & Belabas, N. Quantum state engineering in arrays of nonlinear waveguides. *Physical Review A* **102**, 043706 (2020).
- [54] Serino, L., Gil-Lopez, J., Stefszky, M., Ricken, R., Eigner, C., Brecht, B. & Silberhorn, C. Realization of a multi-output quantum pulse gate for decoding high-dimensional temporal modes of single-photon states. *PRX Quantum* **4**, 020306 (2023).
- [55] Lu, H.-H., Liscidini, M., Gaeta, A. L., Weiner, A. M. & Lukens, J. M. Frequency-bin photonic quantum information. *Optica* **10**, 1655 (2023).
- [56] Oliver, R. et al. N-way frequency beamsplitter for quantum photonics. *arXiv:2405.02453 [quant-ph]* (2024).
- [57] Willner, A. E., Khaleghi, S., Chitgarha, M. R. & Yilmaz, O. F. All-optical signal processing. *Journal of Lightwave Technology* **32**, 660–680 (2013).
- [58] McMahon, P. L. The physics of optical computing. *Nature Reviews Physics* **5**, 717–734 (2023).
- [59] Saxena, M., Eluru, G. & Gorthi, S. S. Structured illumination microscopy. *Advances in Optics and Photonics* **7**, 241 (2015).
- [60] Heist, S., Zhang, C., Reichwald, K., Kühmstedt, P., Notni, G. & Tünnermann, A. 5d hyperspectral imaging: fast and accurate measurement of surface shape and spectral characteristics using structured light. *Optics Express* **26**, 23366 (2018).
- [61] Wang, Z., Li, L., Shi, Y., Chen, J., Yao, J. & Li, Z. Metasurface-empowered five-dimensional imaging with structured light. *ACS Photonics* **11**, 3898–3906 (2024).
- [62] Timurdogan, E., Poulton, C. V., Byrd, M. J. & Watts, M. R. Electric field-induced second-order nonlinear optical effects in silicon waveguides. *Nature Photonics* **11**, 200–206 (2017).
- [63] Chen, P.-K., Briggs, I., Cui, C., Zhang, L., Shah, M. & Fan, L. Adapted poling to break the nonlinear efficiency limit in nanophotonic lithium niobate waveguides. *Nature Nanotechnology* **19**, 44–50 (2023).
- [64] Sylvestre, T. et al. Recent advances in supercontinuum generation in specialty optical fibers [invited]. *Journal of the Optical Society of America B* **38**, F90 (2021).
- [65] Molesky, S., Lin, Z., Piggott, A. Y., Jin, W., Vucković, J. & Rodriguez, A. W. Inverse design in nanophotonics. *Nature Photonics* **12**, 659–670 (2018).
- [66] Zhao, Z. et al. High computational density nanophotonic media for machine learning inference. *arXiv preprint arXiv:2506.14269* (2025).
- [67] Sved, J., Song, S., Li, L., Li, G., Meng, D. & Yi, X. Inverse-designed nanophotonic neural network accelerators for ultra-compact optical computing. *arXiv preprint arXiv:2506.06150* (2025).

- [68] Wu, T., Li, Y., Ge, L. & Feng, L. Field-programmable photonic nonlinearity. *Nature Photonics* 1–8 (2025).
- [69] Xiang, C., Jin, W. & Bowers, J. E. Silicon nitride passive and active photonic integrated circuits: trends and prospects. *Photonics research* **10**, A82–A96 (2022).
- [70] Churaev, M. et al. A heterogeneously integrated lithium niobate-on-silicon nitride photonic platform. *Nature communications* **14**, 3499 (2023).

Authors' response

We would like to thank the reviewers for their thoughtful and detailed evaluations of our work. The reviewers' insightful questions and constructive suggestions have been very helpful in improving our manuscript. Below, we address each comment individually. Reviewer comments are presented *italicized*, modifications to the manuscript in **red**, and deletion from the previous version in **blue**. All substantial changes to the manuscript are listed in the response below unless otherwise noted.

Reviewer #1

The authors have made significant revisions to their paper, following the reviewers comments. These include some clarifications and comparisons to other works, adding some missing data, in particular related to the experimental measurements, report of a new result of channel waveguide and 4 detailed proposals for future applications of this technology. Overall, the authors have made a serious effort to address the reviewers comments and I believe that the manuscript was improved in this process. Perhaps the most important new result is the report on realization and characterization of a channel waveguide. This enabled to address one of the weak points of this new technology - its relatively low nonlinear conversion efficiency. Now that they use channel waveguide, they report a significant increase of 40 times in conversion efficiency. As for the proposals for future applications with this technology, I think that they are interesting, but some of the numbers there are very optimistic. As an example, for Potential Application 1: Highly programmable on-chip optical pulse shaper and for Potential Application 2: Programmable quantum frequency converter, the authors assume that they can maintain the efficiency they obtained for a length of 0.7 mm over a propagation distance of 1.5 meters. It is not clear whether they can maintain the exact phase matching condition over such a long distance. This is a major issue with other, more mature waveguide technologies, for example using thin film LiNbO₃. In addition, the current loss level they reported is 0.4 dB/cm, so it still requires dramatic reduction in the loss values in order to have interaction lengths of meters. Nevertheless, I still think that this paper represents a major new result in nonlinear optics, and I recommend accepting it to Nature.

Reviewer name: Ady Arie

Reply 1.0: We thank the reviewer for the comments and positive evaluation of the work. As pointed out by the reviewer, the potential future applications section involve speculations on device performance beyond what we demonstrated in this work. As shown below, we have added new text to highlight these as essential challenges that future research efforts should be focused on.

Box 1: New text in Supplementary Information

~~For the performance analysis, we take a conservative approach by relying on the nonlinearities and normalized (slope) conversion efficiencies already demonstrated in our work on SiN programmable nonlinear photonics.~~ For the performance analysis of SiN programmable waveguides, we take a

conservative approach to the values of nonlinearities, relying only on normalized (slope) conversion efficiencies already demonstrated in our work. This is because optical nonlinearity is an inherent property of a material and cannot be significantly enhanced even by advanced fabrication techniques unless different materials are employed. On the other hand, propagation loss depends strongly on fabrication processes, and we expect there is more room for improvement beyond the values demonstrated in the prototype devices. Thus, we cite loss numbers reported in the literature for the performance estimations. Advances in fabrication techniques to reduce propagation loss in programmable waveguides would be an essential step to realize some of the applications proposed in this section. Finally, some of the applications require QPM gratings over a meter in length, which is well beyond the centimeter scale demonstrated in this work. Developing a scheme to efficiently optimize the QPM grating structure based on real-time feedback would be essential to realize such operations.

Reviewer #2

I co-reviewed this manuscript with one of the reviewers who provided the listed reports.

Reply 2.0: We thank the reviewer for providing a review of our manuscript. Please refer to the corresponding section of this document for our response to the comments.

Reviewer #3

In my previous review, I noted the creativity and quality of the results presented in this manuscript but had concerns about whether the low conversion efficiency would be prohibitive for practical applications, and also wanted more clarity as to envisioned applications of the authors' programmable nonlinear waveguide approach. I am pleased to say that in their revised manuscript, supplementary material, and response letter, the authors have done a thorough job of addressing both concerns. I therefore endorse publication in Nature.

For the former, they have first carefully outlined sources of inefficiency, focusing on how their effective second-order nonlinearity is actually quite significant (e.g., on par or larger than the highest demonstrated in SiN) and it is mostly their specific device geometry and limitations with loss and modal confinement that prevent higher efficiency from being obtained. They then conducted new experiments in a channel waveguide geometry, introducing lateral confinement that is absent in the prior slab waveguide results, and show that this improves the SHG efficiency by about a factor of 40, making it about one order of magnitude lower than prior results in channel waveguides that exhibit a greater degree of confinement (in those works, the effective $\chi^{(2)}$ is photoinduced). Given that there are many possible ways to further engineer the system, in terms of stronger optical mode confinement, reduced optical loss, resonant enhancement, etc, and none of these are fundamentally incompatible with the authors' approach (albeit there will likely be tradeoffs), I believe their demonstrations have now answered the basic question of whether useful conversion efficiencies can be realized with their approach.

With regards to the latter, the authors have presented four future applications in the supplementary material (summarized in the main text), all of which I believe are credible and point to what may be achievable with some additional levels of device improvement.

Reply 3.0: We thank the reviewer for the comments and recognition of our contribution.

Reviewer Comment 3.1 — *My remaining comments are below: 1. The authors could consider including Table R1 in their supplementary material. This clear comparison against the prior literature can be helpful for readers. They could potentially additionally include a ‘bandwidth’ column since conversion efficiency at a single frequency isn’t the only metric in many important cases.*

Reply 3.1: We thank the reviewer for the helpful suggestion. We have included the table summarizing the performance of prior literature, with a new column for “bandwidth”, in Supplementary Information. The newly added text is shown below.

Box 2: New table and text in Supplementary Information

Device type	Induced $\chi^{(2)}$ nonlinearity	Conversion efficiency η_{norm}	Bandwidth	Reference
Programmable planar waveguide	0.47 pm/V	$5 \times 10^{-5} \%$ /W	700 GHz ^a	This work
Programmable channel waveguide	0.47 pm/V	$2 \times 10^{-3} \%$ /W	700 GHz	This work
Channel waveguide	0.5 pm/V	$5 \times 10^{-3} \%$ /W	70 THz ^b	Ref. [1]
Channel waveguide	0.3 pm/V	$5 \times 10^{-2} \%$ /W	300 GHz ^c	Ref. [2]
Microring resonator	0.03 pm/V	47.6 %/W	260 MHz ^d	Ref. [3]
Microring resonator	0.2 pm/V	2500 %/W	320 MHz ^e	Ref. [4]
Microring resonator	Not provided	651 %/W	14 MHz ^f	Ref. [5]
Microring resonator	0.03 pm/V	3.2 %/W	350 MHz ^g	Ref. [6]
Microring resonator	0.022 pm/V	141 %/W	13 MHz ^h	Ref. [7]

Table S1: Summary of the performance of electric-field induced $\chi^{(2)}$ nonlinear-optical device on SiN. For waveguides, the bandwidth is defined as the width of the SHG phase-matching function. For resonators, we refer to the linewidth of the fundamental harmonics. ^a 5.6 nm around the pump wavelength of 1562 nm. ^b 60 nm around the pump wavelength of 1560 nm. ^c 2.5 nm around the pump wavelength of 1544 nm. ^d $Q = 7.5 \times 10^5$ around 1550 nm. ^e $Q = 6.0 \times 10^5$ around 1560 nm. ^f $Q = 1.39 \times 10^7$ around 1560 nm. ^g $Q = 8 \times 10^6$ around 1060 nm. ^h $Q = 2.2 \times 10^7$ around 1063 nm.

To put the performance of the programmable nonlinear waveguides into context, in Table S1 we compare the performance of our devices with literature values reporting electric-field-induced second-harmonic generation (SHG) in SiN nanophotonics. In all cases except our work, the $\chi^{(2)}$ nonlinearity arose from the photogalvanic effect. As shown in the table, the programmable $\chi^{(2)}$ nonlinearity demonstrated in our work is comparable to, or even greater than, previously reported values. The apparently lower conversion efficiencies are primarily due to differences in device geometries, such as shorter propagation lengths and weaker transverse field confinement. As can be seen in the table, using a resonant structure can significantly enhance the conversion efficiency by recirculating both the pump and the generated SH light, at the cost of the bandwidth. As

discussed in Sec. S2 C, forming a microring resonator with our programmable channel waveguide could achieve performance surpassing the current state of the art in nanophotonics, albeit with technical challenges in reducing propagation loss.

Reviewer Comment 3.2 — 2. *I didn't really understand what I was supposed to be looking at in Figure 6(b) – is the channel waveguide visible in this optical image, or is the main point to show the waveguide length? What is the rectangular region within the chip denoting?*

Reply 3.2: We thank the reviewer for the comment. Figure 6(b) is an image of the programmable planar waveguide, and thus, there is no fine channel waveguide structures visible in the image. The rectangular region is the edge of the transparent electrodes. To make these points clear, we have revised the figure captions and relevant text as shown below.

Box 3: New caption for the image of the programmable planar waveguide

(a) Illustration of the stack structure of a programmable nonlinear **planar** waveguide. (b) A programmable nonlinear **planar** waveguide that was fabricated and used to produce parts of the results in the main text. **The rectangular line visible on the surface of the chip is the edge of the transparent electrodes.**

Reviewer Comment 3.3 — 3. *In the channel waveguide experiments, did the authors observe any SHG in absence of an explicit applied DC field (i.e., do they observe any SHG mediated by the photogalvanic effect for DC field creation)? It could be interesting to compare the results of a photo-induced SHG vs. SHG via an explicit DC field if that is the case.*

Reply 3.3: We thank the reviewer for the comment. In our experiment on the programmable channel waveguide, we did not observe obvious contributions from the photogalvanic effect. We attribute this to the weak power of the pump light; Photogalvanic effect is induced by the nonlinear polarization caused by the pump field E_ω and SH field $E_{2\omega}$, taking a form $\propto E_{2\omega}^* E_\omega^2$. In order for such nonlinear polarization to cause strong enough induced $\chi^{(2)}$, both the pump and the SH fields have to be strong. To achieve this, previous demonstrations utilized high-Q resonators or strong pump light to enhance the intensity of the optical fields. For instance, Ref. [2] used 90 W of peak pump power to activate the photogalvanic effect on a waveguide. On the other hand, in our experiment on the channel waveguide, we used a CW pump laser with less than 5 mW of on-chip power. Combined with the low transverse field confinement of the waveguide, we expect the nonlinear polarization could not develop to the extent of inducing measurable $\chi^{(2)}$.

Reviewer Comment 3.4 — 4. *Do the authors have any ideas about the fundamental and second harmonic mode identities (e.g., polarization and transverse profile) in the channel waveguide experiments? If so, it would be useful to note this and potentially including simulated field profiles in Fig. 7. Regarding polarization, it would seem that their DC field is vertically oriented – does that mean they focus primarily on TM-polarized modes?*

Reply 3.4: We thank the reviewer for the comment. The comment has helped us recognize that we were not very clear about the nature of the modes used in the channel waveguide experiments. In our experiment, we coupled light polarized in the vertical (i.e., y) direction to the waveguide, which is

expected to excite the fundamental transverse magnetic (TM) mode of the pump wavelength. The DC field is also oriented in the vertical direction, which induces vertical $\chi^{(2)}$ nonlinearity, leading to the SHG into the fundamental TM mode of the SH wavelength. To make this clear, we have added the clarifying remarks and added simulated field profiles of the waveguide modes in Fig. 7, as shown below.

Box 4: New text in Methods

~~We coupled CW pump light with various wavelengths between 1540 nm and 1600 nm and measured the generated SHG while scanning the period of the monotonic QPM grating.~~ We coupled CW pump light with various wavelengths between 1540 nm and 1600 nm to the waveguide. The pump light is polarized in the vertical (i.e., y) direction, which is expected to mainly excite the fundamental TM mode, and the SHG takes place to the fundamental TM mode of the SH wavelength. We show the mode profiles of these modes in Extended Data Fig. 2(b). We measured the generated SHG while scanning the period of the monotonic QPM grating.

Box 5: New figure in Methods

Extended Data Fig. 2 (a) An SEM image of the cross section of a programmable nonlinear channel waveguide. We highlight the boundaries between different materials with dashed lines. The white scale bar represents 2 μm . (b) Numerically simulated distribution of the vertical electric field E_y of the fundamental transverse-magnetic (TM) mode at the wavelength of 1560 nm (left) and 780 nm (right). The gray scale bar represents 1 μm . The simulation was performed with EMpy [8].

Reviewer Comment 3.5 — 5. I found the description of the fabrication process for the channel waveguides to be a little unclear in a couple of places:

a. They mention first depositing 500 nm thick SiO₂ on the wafer they ordered (which had a lower SiO₂ cladding a SiN core), then performing photolithography and pattern transfer to a Cr

hard mask, and then etching through the SiN layer. Just to confirm, aren't they etching through both the 500 nm SiO₂ as well as the 2 μm SiN in this step?

b. They mention then depositing an additional 3 μm of SiO₂ before etching away 2.5 μm of it. One may wonder why they didn't simply deposit 500 nm of additional SiO₂... I assume they followed the approach mentioned to planarize the waveguide? A comment may be worth including if this is the case.

c. The text states that they deposit a 12 μm SRN layer but the Fig. 6(a) shows this layer as being 7.5 μm thick.

Reply 3.5: Regarding point (a): It is correct that this etching process etched through both the 500 nm of SiO₂ layer and the SiN layer. We added new text to clarify on this point. Also, we have corrected a typo in the gas composition.

Regarding point (b): This process was performed to make sure that the channel waveguide has enough cladding covering the sidewalls. We added clarification on this point in Methods as shown below.

Regarding point (c): The programmable planar waveguide and programmable channel waveguide we fabricated in this study had different thickness of the SRN layer, i.e., 7.5 μm for the former and 12 μm for the latter. Since Fig. 6(a) represents the stack structure of the programmable planar waveguide, the SRN thickness is supposed to be 7.5 μm. We added new text to the figure caption to clarify that the figure represents the structure of the programmable planar waveguide.

Box 6: New text in Methods

We then etched through ~~the SiN layer using CHF₃/O₂ gases~~ the SiO₂ and SiN layer using CHF₃/O₂/N₂ gases in a plasma etcher (PlasmaPro 100 RIE; Oxford Instruments).

Box 7: New text in Methods

~~To deposit additional cladding on the SiN sidewalls and thereby reduce losses into the photoconductive layer, we deposited 3 μm of SiO₂ with PECVD, and subsequently etched away 2.5 μm of oxide using the same plasma etching recipe described above.~~ We then deposited 3 μm of SiO₂ with PECVD and subsequently etched away 2.5 μm of oxide using CHF₃/O₂ gases in the same plasma etcher, which added 500 nm of oxide on top of the waveguide. Importantly, 1.5 μm of oxide remained on the waveguide sidewalls due to the conformal oxide deposition and anisotropic oxide etching. This extra oxide on the sidewall reduces loss into the photoconductor layer.

Reviewer #4

The authors have done an excellent job addressing the reviewers' comments. The revisions have significantly strengthened the manuscript. The expanded functionalities and capabilities demonstrated here surpass prior work, but also highlight the novelty and impact of this study. I am pleased to recommend this paper for publication in its present form.

Reply 4.0: We thank the reviewer for the comment and positive evaluation of our work.

Additional edits

References

- [1] Hickstein, D. D. et al. Self-organized nonlinear gratings for ultrafast nanophotonics. *Nature Photonics* **13**, 494–499 (2019).
- [2] Billat, A., Grassani, D., Pfeiffer, M. H. P., Kharitonov, S., Kippenberg, T. J. & Brès, C.-S. Large second harmonic generation enhancement in Si₃N₄ waveguides by all-optically induced quasi-phase-matching. *Nature Communications* **8** (2017).
- [3] Nitiss, E., Hu, J., Stroganov, A. & Brès, C.-S. Optically reconfigurable quasi-phase-matching in silicon nitride microresonators. *Nature Photonics* **16**, 134–141 (2022).
- [4] Lu, X., Moille, G., Rao, A., Westly, D. A. & Srinivasan, K. Efficient photoinduced second-harmonic generation in silicon nitride photonics. *Nature Photonics* **15**, 131–136 (2020).
- [5] Li, B. et al. Down-converted photon pairs in a high-Q silicon nitride microresonator. *Nature* **639**, 922–927 (2025).
- [6] Wang, G. et al. Integrated tunable green light source on silicon nitride. *arXiv preprint arXiv:2504.13662* .
- [7] Yuan, Z. et al. Efficient and wavelength-tunable second-harmonic generation toward the green gap. *Science Advances* **11**, eadw2781 (2025).
- [8] Bolla, L. EMpy: Electromagnetic Python. <https://github.com/lbolla/EMpy> (2017).